# Tropomyosin 1-I/C coordinates kinesin-1 and dynein motors during *oskar* mRNA transport

Simone Heber [1,7], Mark A. McClintock[2,7], Bernd Simon[3,4], Eve Mehtab [1], Karine Lapouge[5], Janosch Hennig [3,6], Simon L. Bullock [2] ✉ & Anne Ephrussi [1] ✉

Dynein and kinesin motors mediate long-range intracellular transport, translocating towards microtubule minus and plus ends, respectively. Cargoes often undergo bidirectional transport by binding to both motors simultaneously. However, it is not known how motor activities are coordinated in such circumstances. In the *Drosophila* female germline, sequential activities of the dynein–dynactin–BicD–Egalitarian (DDBE) complex and of kinesin-1 deliver *oskar* messenger RNA from nurse cells to the oocyte, and within the oocyte to the posterior pole. We show through in vitro reconstitution that Tm1-I/C, a tropomyosin-1 isoform, links kinesin-1 in a strongly inhibited state to DDBE-associated *oskar* mRNA. Nuclear magnetic resonance spectroscopy, small-angle X-ray scattering and structural modeling indicate that Tm1-I/C suppresses kinesin-1 activity by stabilizing its autoinhibited conformation, thus preventing competition with dynein until kinesin-1 is activated in the oocyte. Our work reveals a new strategy for ensuring sequential activity of microtubule motors.

Motor-driven transport along microtubules is critical for the localization of many cellular constituents. Dynein and kinesin motors translocate towards microtubule minus ends and plus ends, respectively. However, their cargoes often travel bidirectionally because of a stable association with both motor classes[1,2]. The distribution of such cargoes in the cell is determined by the relative activities of the bound dyneins and kinesins but how this is orchestrated at the molecular level is unclear.

Individual motors appear to be in an autoinhibited state by default, with activation driven by adaptor proteins and cargoes. Dynein motility is switched on by the dynactin complex and one of a number of coiled coil-containing activating adaptors[3–7]. The prototypical kinesin motor, kinesin-1, is a tetramer of two kinesin heavy chains (Khc) and two kinesin light chains (Klc)[8]. Khc contains an N-terminal globular motor domain, a coiled-coil stalk domain and a C-terminal disordered tail domain (Fig. 1a), and is autoinhibited by docking of a conserved isoleucine–alanine–lysine (IAK) motif in the tail on the motor domain[9–12]. The autoinhibited conformation is stabilized by Klc[13,14], which binds to the Khc stalk[15,16] and contributes to cargo recognition[17]. The roles these interactions play in controlling dynein and kinesin activity during bidirectional transport are not understood.

*oskar* (*osk*) mRNA localization in the *Drosophila* egg chamber is an attractive system for studying dual motor transport. Delivery of *osk* to the posterior pole of the developing oocyte, which drives abdominal patterning and germline formation in the embryo[18], is driven by the successive activities of dynein[19] and kinesin-1 (refs. 20,21). In early oogenesis, *osk* mRNA that is synthesized in the nurse cells is transported into the interconnected oocyte by dynein in complex with dynactin and the activating adaptor Bicaudal D (BicD), which is linked to double-stranded mRNA localization signals by the RNA-binding protein Egalitarian (Egl)[19,22–24]. Association of Egl with BicD and consequent dynein activation are enhanced by binding of Egl to RNA[25,26],

[1]Developmental Biology Unit, European Molecular Biology Laboratory, Heidelberg, Germany. [2]Division of Cell Biology, MRC Laboratory of Molecular Biology, Cambridge, UK. [3]Structural and Computational Biology Unit, European Molecular Biology Laboratory, Heidelberg, Germany. [4]Department of Molecular Biology and Biophysics, University of Connecticut Health Center, Farmington, CT, USA. [5]Protein Expression and Purification Core Facility, European Molecular Biology Laboratory, Heidelberg, Germany. [6]Biochemistry IV, Biophysical Chemistry, University of Bayreuth, Bayreuth, Germany. [7]These authors contributed equally: Simone Heber, Mark A. McClintock. ✉e-mail: sbullock@mrc-lmb.cam.ac.uk; anne.ephrussi@embl.org

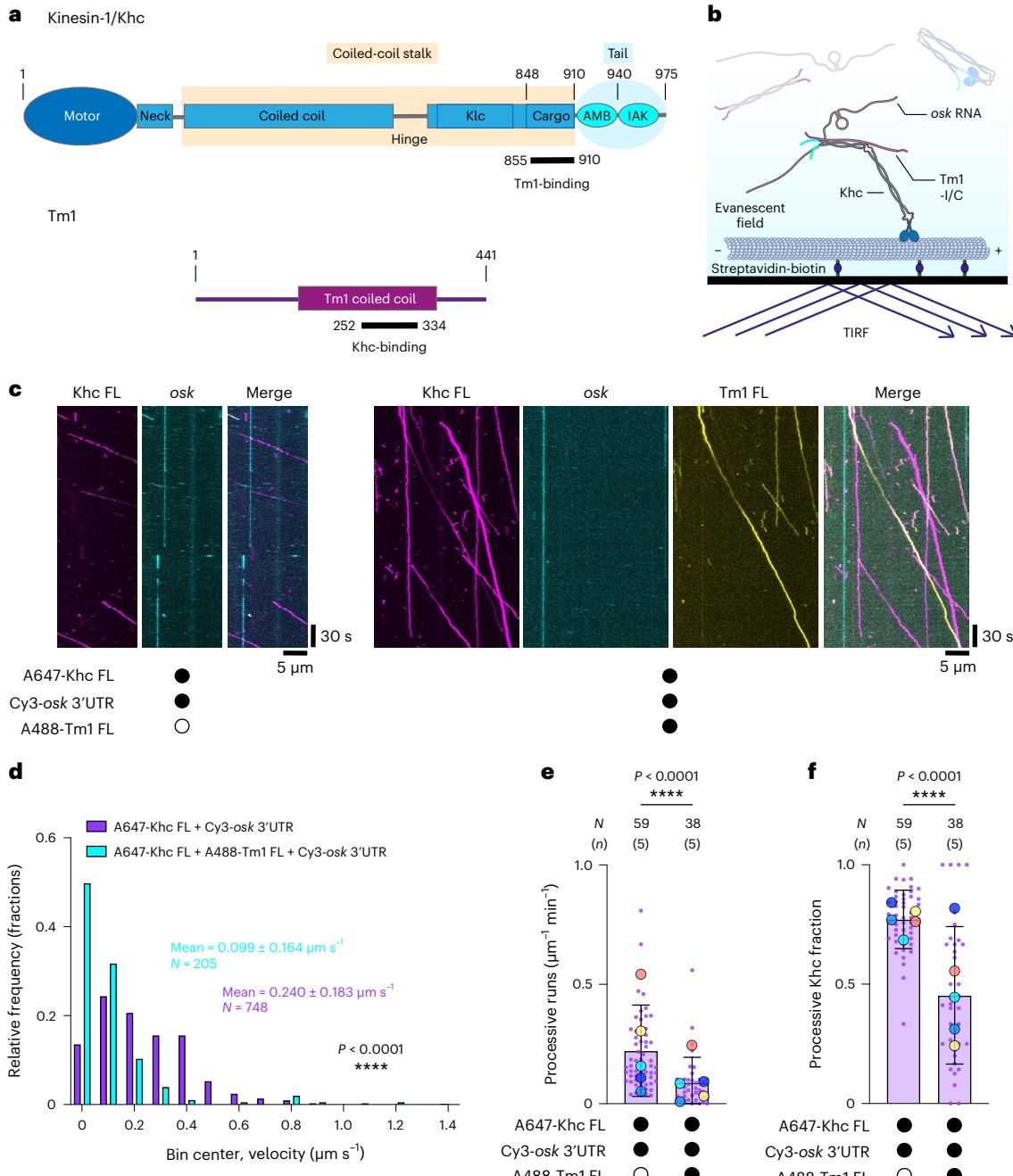

**Fig. 1 | In vitro reconstitution reveals an inhibitory effect of Tm1 on Khc motility. a**, Domain architecture of *Drosophila* Khc and atypical Tm1. In isolation, Khc and Tm1 form coiled-coil homodimers (parallel and antiparallel, respectively). When in complex, Tm1 and Khc form a trimeric coiled coil comprising two parallel Khc chains and one antiparallel Tm1 chain[32]. Boxed regions of Khc represent binding sites for Klc or cargo. **b**, Scheme of TIRF-based in vitro motility assay. **c**, Kymographs (projecting the position of fluorescently labeled particles over time along an individual microtubule; moving particles have a horizontal component, whereas static particles are projected as vertical lines) showing the motile behavior of Alexa Fluor 647 (A647)-labeled Khc FL (labeling efficiency: 79.75%) and Cy3-labeled *osk* 3′ UTR (labeling efficiency ~100%) with or without Alexa Fluor 488 (A488)-labeled Tm1 FL (labeling efficiency: 57%). Microtubule plus and minus ends are oriented toward the right and left of each kymograph, respectively. **d**, Velocity distributions of Khc FL

with or without Tm1 FL. Mean ± s.d. is indicated (for breakdown of means across independent experimental replicates, see Fig. 2a). **e,f**, Frequency of processive Khc FL (**e**) and fraction of Khc FL binding events that underwent processive motility (**f**) with or without Tm1 FL. Mean ± s.d. is shown and is derived from 38–59 microtubules (*N*), representing analysis of 205–748 individual complexes from 10–15 imaging chambers and 5 independent experiments (*n*) per condition (Supplementary Table 1). Mean values for each independent experiment (large dots) are superimposed on values for individual microtubules (small dots). In all panels, black or white circles indicate the presence or absence of the indicated components, respectively. Parallel experiments within the figure are shown with large dots of the same color. Cy3-*osk* 3′ UTR was included in all experiments. Statistical significance was determined by unpaired two-tailed *t*-tests with Welch's correction using *N* values (total number of individual complexes (**d**) or total number of microtubules (**e,f**)).

indicating a role for the cargo in promoting dynein activity. In early oogenesis, microtubule minus ends are nucleated in the oocyte, consistent with the dynein-based delivery of mRNAs into this cell. During

mid-oogenesis, the polarity of the microtubule network shifts dramatically, with plus ends pointing towards the oocyte posterior. At this stage, Khc translocates *osk* to the posterior pole[27]. This process is

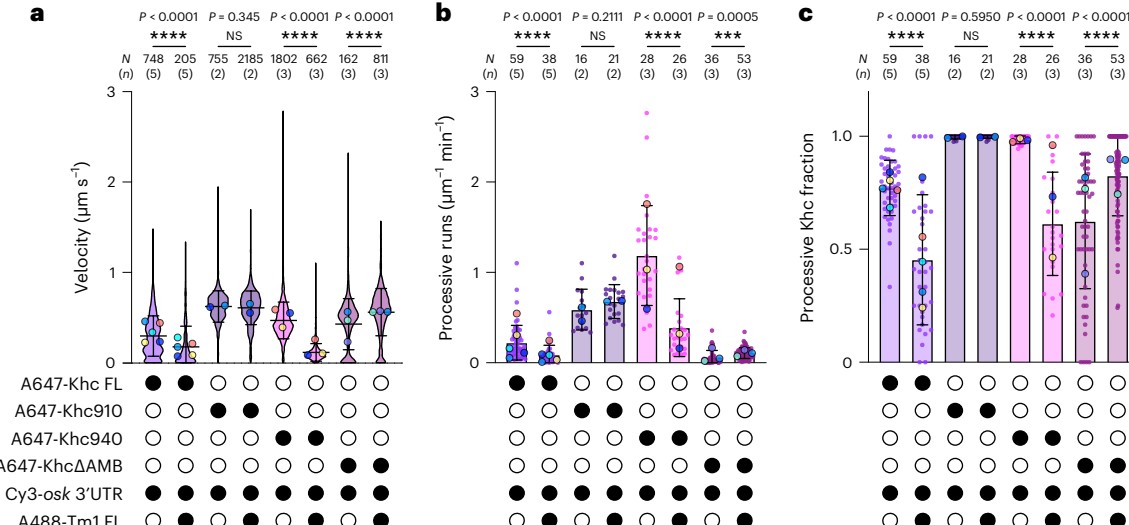

**Fig. 2 | Tm1 inhibits Khc motility via Khc's regulatory tail. a–c**, Velocity of motile Khc complexes (**a**), frequency of processive Khc movements (**b**) and fraction of Khc microtubule-binding events that underwent processive motility (**c**) for Khc FL, Khc910, Khc940 and KhcΔAMB with or without Tm1 FL. Labeling efficiencies for Khc FL, Khc910, Khc940 and KhcΔAMB were 79.8%, 54%, 78% and 78.8%, respectively, so that total run numbers are underestimated. For all plots, the mean ± s.d. is shown and is derived from 162–2,185 individual complexes (**a**; *N*) or 16–59 microtubules (**b**,**c**; *N*) from 4–15 imaging chambers and 2–5 independent experiments (*n*) per condition (Supplementary Table 1). Mean values for each independent experiment (large dots) are superimposed on a violin plot showing the distribution of values from individual complexes (**a**) or on values for individual microtubules (small dots; **b**,**c**). Black or white circles indicate the presence or absence of the indicated components, respectively. Parallel experiments within the figure are shown with large dots of the same color. Cy3-*osk* 3′ UTR was included in all experiments. Statistical significance was determined by unpaired two-tailed Mann–Whitney test (**a**) or unpaired two-tailed *t*-tests with Welch's correction (**b**,**c**) using *N* values (total number of individual complexes (**a**) or total number of microtubules (**b**,**c**)). NS, not significant (*P* > 0.05). Although Tm1 significantly increased the metrics of KhcΔAMB activity when comparing data that were aggregated between independent replicates, consistent effects were not seen between experiments. Data for Khc FL are reproduced from Fig. 1d–f.

independent of Klc[28,29], raising the question of how Khc is linked to *osk* and how its motor activity is regulated.

Transport of *osk* RNA by Khc requires the unique I/C isoform of tropomyosin-1, Tm1-I/C (hereafter Tm1)[30,31]. Tm1 binds to a noncanonical but conserved cargo-binding region in the Khc tail and stabilizes interaction of the motor with RNA, suggesting a function as an adaptor[32]. Both Khc and Tm1 are loaded onto *osk* ribonucleoprotein particles (RNPs) shortly after their export from the nurse cell nuclei, although the motor only appears to become active in the mid-oogenesis oocyte[31]. Similarly, dynein remains associated with *osk* RNPs during Khc-mediated transport within the oocyte, but is inactivated by displacement of Egl by Staufen[33,34]. How the two motors are linked simultaneously to *osk* RNPs, and how Khc is inhibited during dynein-mediated transport into the oocyte, is not known.

Here, we show that Tm1 inhibits Khc by stabilizing its autoinhibited conformation through a new mechanism involving the motor's regulatory tail domain and stalk. Tm1 also links Khc to the dynein-transported *osk* RNP, thereby allowing cotransport of inactive Khc on *osk* RNA by dynein. In vivo, such a mechanism would avoid competition between the two motors during delivery of *osk* RNPs to the oocyte by dynein, while ensuring that Khc is available on these structures to mediate their delivery to the oocyte posterior in mid-oogenesis. With its cargo-binding and motor regulatory functions, we propose that Tm1 is a noncanonical light chain for kinesin-1.

## Results

### Khc transports RNA and is inhibited by Tm1

To test the ability of Khc and the Khc–Tm1 complex to transport *osk* RNA along microtubules, we performed single-molecule resolution in vitro motility assays with recombinant proteins (Fig. 1a,b)[25]. In the absence of Tm1, we observed occasional comigration of the *osk* 3′ untranslated region (UTR) with Khc (47 of 748 Khc complexes (6.3%);

Fig. 1c). This frequency did not change significantly in the presence of Tm1 (9 of 205 Khc complexes (4.2%); *P* = 0.4016, Fisher's exact test), despite 44.6% of Khc signals colocalizing with Tm1 signals. Correcting for the efficiency of fluorophore labeling of Tm1 (57%) indicated that ~95% of Khc molecules were bound to Tm1. The high occupancy of Khc with Tm1 is consistent with the strong binding observed previously with these proteins[32]. Mass photometry confirmed high occupancy of Khc with Tm1 at nanomolar concentrations (Supplementary Fig. 1), as well as the proposed 2:1 stoichiometry of the Khc–Tm1 complex[32]. We conclude from these experiments that Khc can associate with *osk* and that the Khc–Tm1 complex does not have substantially higher affinity for the RNA in these assay conditions.

We next examined the effect of Tm1 on Khc motility. Without Tm1 present in the assay, microtubule-associated Khc complexes moved with a mean velocity of 0.240 μm s⁻¹ (Fig. 1d). In the presence of Tm1, the mean velocity of Khc decreased to 0.099 μm s⁻¹ (Fig. 1d). Inclusion of Tm1 also reduced the frequency of processive Khc movement (Fig. 1e) and microtubule-binding events (Extended Data Fig. 1), as well as the fraction of Khc microtubule-binding events that resulted in processive motility (Fig. 1f). These data reveal an inhibitory effect of Tm1 on Khc motility.

In addition to the 3′ UTR, an RNA structure in the *osk* coding sequence—the spliced *oskar* localization element (SOLE)—is important for Khc-mediated transport of *osk* to the oocyte posterior[35,36]. We tested whether an RNA including the SOLE could stimulate the motility of Khc in the presence of Tm1 by replacing the *osk* 3′ UTR in our assay with a 135-nucleotide RNA (*osk* min) comprising the SOLE and the oocyte entry signal of *osk* (Supplementary Fig. 2a), which resides in the 3′ UTR and promotes dynein-mediated transport[37]. Khc motility was also impaired by Tm1 in the presence of *osk* min (Supplementary Fig. 2b–d). Thus, Tm1 inhibits motility of Khc in both the presence and absence of the SOLE.

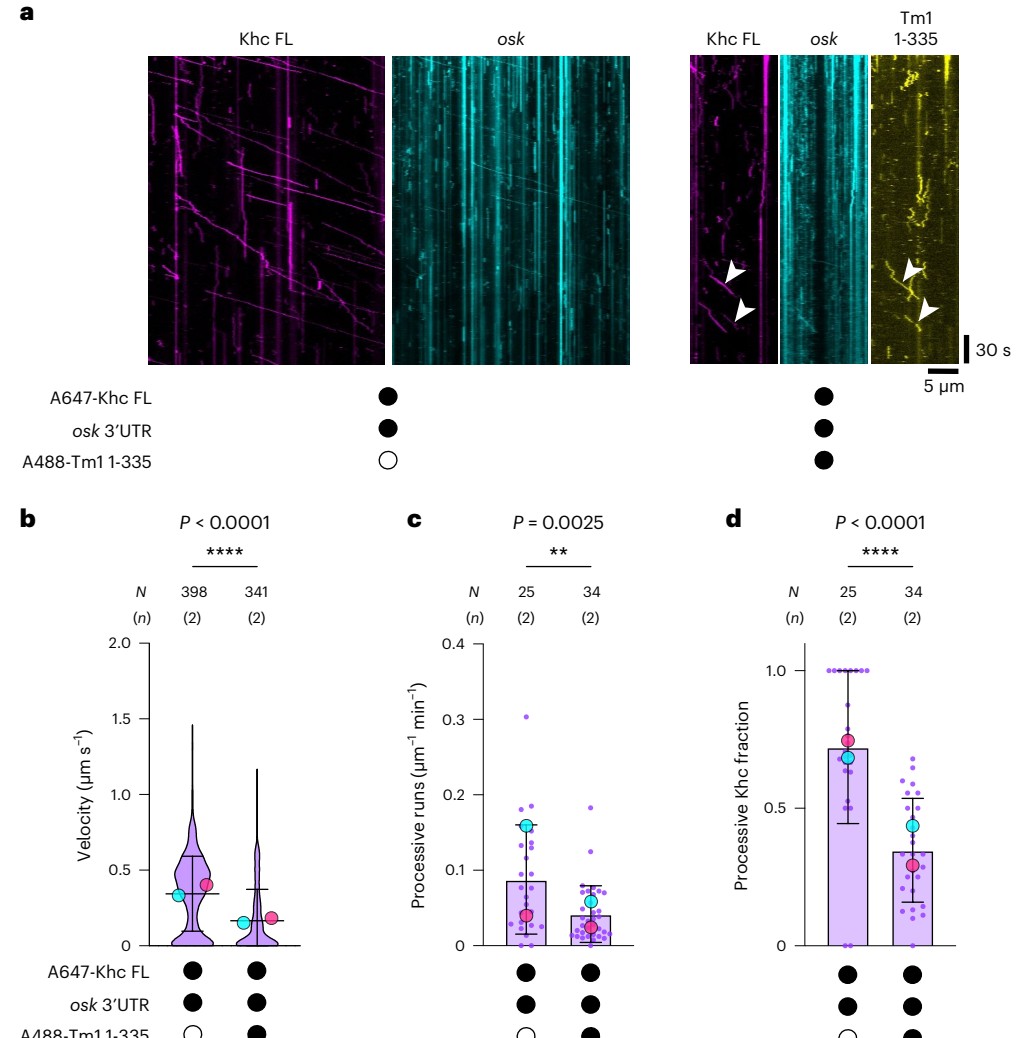

**Fig. 3 | The N-terminal region of Tm1 is sufficient for Khc inhibition.**
**a**, Kymographs showing motile behavior of Khc FL and *osk* 3′ UTR with or without Tm1 1–335. Microtubule plus and minus ends are oriented toward the right and left of each kymograph, respectively. Khc movements that colocalize with Tm1 are indicated by white arrowheads. **b–d**, Velocity of motile Khc FL (**b**), frequency of processive Khc FL movements (**c**) and fraction of Khc FL binding events that underwent processive motility (**d**) in the presence or absence of Tm1 1–335. For all plots, mean ± s.d. is shown and is derived from 341–398 individual complexes (**b**; *N*) or 25–34 microtubules (**c,d**; *N*) from 4–6 imaging chambers and 2 independent experiments (*n*) per condition (Supplementary Table 2). Mean values for each independent experiment (large dots) are superimposed on a

violin plot showing the distribution of values from individual complexes (**b**) or on values for individual microtubules (small dots; **c,d**). Black or white circles indicate the presence or absence of the indicated components, respectively. Parallel experiments within the figure are shown by large dots of the same color. Cy3-labeled or unlabeled *osk* 3′ UTR was included in all experiments. Statistical significance was determined by unpaired two-tailed Mann–Whitney test (**b**) or unpaired two-tailed *t*-tests with Welch's correction (**c,d**) using *N* values (total number of individual complexes (**b**) or total number of microtubules (**c,d**)). The experiment shown in cyan also compared Khc FL metrics with or without Tm1 FL (data set also incorporated in Figs. 1d–f and 2).

## Tm1 inhibits Khc motility via the Khc regulatory tail

We next set out to identify the region(s) of Khc involved in inhibition by Tm1. Regulatory functions have been attributed to an auxiliary, ATP-independent microtubule-binding (AMB) domain and the auto-inhibitory IAK motif within the Khc tail (Fig. 1a)[11,38,39]. Whereas the AMB domain promotes microtubule-sliding by Khc[40], the IAK motif inhibits Khc's ATPase activity[9,11], and thus processive motility, by binding to the Khc motor domain and thereby preventing neck–linker undocking and ADP release[10]. We tested whether the tail is responsible for Tm1-mediated inhibition of Khc by assessing the effect of Tm1 on the motility of a Khc truncation lacking both the AMB domain and the IAK motif (Khc 1–910 (hereafter Khc910); Fig. 1a). Consistent with the removal of autoinhibitory elements, Khc910 was more active than full-length Khc (Khc FL), exhibiting increased velocity, as well as more frequent processive runs and microtubule-binding events, and an

increased likelihood of processive movement upon microtubule binding (Fig. 2a–c, Extended Data Figs. 1 and 2a and Supplementary Table 1). As with Khc FL, Tm1 did not affect the ability of Khc910 to interact with *osk* RNA (Extended Data Fig. 3a). Unlike Khc FL, however, Khc910 motility was not impaired by Tm1 (Fig. 2a–c, Extended Data Fig. 2a and Supplementary Table 1). Thus, the Khc tail mediates inhibition by Tm1.

We next assessed the requirement of the IAK motif for Tm1-mediated inhibition by truncating Khc just after the AMB domain and just before the IAK motif (Khc940). In keeping with the role of the IAK in Khc inhibition, Khc940 exhibited an increase in velocity, microtubule-binding events, processive runs and the fraction of microtubule-binding events that resulted in processive motility relative to Khc FL (Fig. 2a–c, Extended Data Figs. 1 and 2b and Supplementary Table 1). Addition of Tm1 to Khc940 consistently decreased each of these metrics (Fig. 2a–c, Extended Data Fig. 2b and Supplementary

Table 1). These observations show that inhibition of Khc motility by Tm1 is not dependent on the IAK motif.

### Khc inhibition by Tm1 is mediated by the AMB domain

To test whether the AMB domain in the Khc tail is required for inhibition by Tm1, we produced a Khc mutant that lacks only the AMB domain (amino acids 911–938; KhcΔAMB). Relative to Khc FL, KhcΔAMB did not consistently increase velocity, frequency of processive movement, microtubule binding or the likelihood of processive movement upon microtubule binding (Fig. 2a–c, Extended Data Figs. 1 and 2c and Supplementary Table 1). These observations indicate that autoinhibition is not disrupted in the absence of the AMB. Strikingly, despite the continued ability of KhcΔAMB to associate with both Tm1 and *osk* RNA (Extended Data Fig. 3a,b), the velocity, frequency, microtubule-binding activity and likelihood of processive KhcΔAMB movements were not decreased by Tm1 (Fig. 2a–c, Extended Data Figs. 1 and 2c and Supplementary Table 1). We conclude that the AMB domain of Khc is essential for inhibition by Tm1 but not for Tm1 binding.

### The N-terminal region of Tm1 mediates Khc inhibition

We next investigated which features of Tm1 are responsible for Khc inhibition. We previously showed that a truncated Tm1 protein (amino acids 1–335 of the 441-amino acid full-length protein (Tm1 1–335)) comprising the disordered N-terminal domain and the minimal Khc-binding region of the coiled-coil domain (Fig. 1a) is sufficient for *osk* RNA localization in vivo[32]. In our in vitro motility assay, Tm1 1–335 retained its ability to bind Khc FL (71% of Khc signals colocalized with Tm1 signals; 242 of 341 complexes) and strongly inhibited its motility (Fig. 3a–d and Supplementary Table 2). These data show that the N-terminal region of Tm1 is sufficient to inhibit Khc.

### Tm1 binding affects the disordered region of the Khc tail

Because the inhibitory effect of Tm1 is mediated through the AMB domain, we hypothesized that Tm1 alters microtubule binding by the Khc tail. To test this, we performed microtubule-pelleting assays with the recombinant Khc tail and Tm1. As a positive control, we confirmed that a fraction of both Khc FL and Khc tail (Khc 855–975) pelleted with microtubules (Extended Data Fig. 4a,b). A small fraction of Tm1 1–335 also associated with microtubules. However, we did not observe increased binding of the Khc tail to microtubules in the presence of Tm1 1–335 (Extended Data Fig. 4b). Thus, association with Tm1 does not stimulate microtubule binding of the Khc tail.

In the crystal structure of the Khc tail bound to Tm1, the Khc coiled coil extends into the AMB domain (last visible amino acid: 923)[32]. Attempts to crystallize the same Khc protein alone were unsuccessful, raising the possibility that it is more flexible in the absence of Tm1. It is conceivable that the extended coiled coil of Khc is induced upon Tm1 binding, which could reduce the flexibility of the tail, potentially locking it in a conformation that impairs Khc activity. To test this hypothesis, we probed the solution structure of the Khc tail (Khc 855–975) by nuclear magnetic resonance (NMR) spectroscopy. The $^1$H-$^{15}$N HSQC NMR spectrum of the Khc tail showed 34 resonances, each corresponding to a backbone N-H correlation (Fig. 4a). Backbone assignment experiments and the resulting secondary chemical shifts revealed that the visible peaks are disordered and correspond to amino acids 937–975, the part of the Khc tail containing the IAK motif (Supplementary Fig. 3). Resonances for amino acids 855–936 were broadened beyond

detection because of short transverse relaxation times. This observation is consistent with this part of the protein forming an extended coiled coil, because these structures exhibit slow rotational tumbling of the axis perpendicular to the helix orientation[41]. Circular dichroism (CD) measurements for Khc 855–975 produced a signal with a 222/208 nm molar residue ellipticity ratio of 0.97, consistent with extensive coiled-coil sequence[42] (Supplementary Fig. 4). Collectively, these data suggest that the region containing the AMB domain is at least partly coiled coil in nature even in the absence of Tm1.

We next determined $^1$H-$^{15}$N HSQC spectra for Khc 855–975 following titration of increasing amounts of unlabeled Tm1 1–335. Tm1 decreased the intensity of resonances corresponding to several residues in the IAK and flanking regions (Fig. 4b,c), indicating changes to their chemical environment. This effect could be observed both in the presence and absence of a U10 RNA (Fig. 4c,d), and depended on the N-terminal domain of Tm1 because a Tm1 construct containing only the coiled-coil region (amino acids 252–334) did not affect the spectra (Fig. 4e). These findings are in agreement with our observations that Tm1 inhibits Khc motility despite infrequent RNA binding (Extended Data Fig. 3a) and that the N-terminal region of Tm1 is required for this effect (Fig. 3). In the presence of Tm1 1–335, the intensities of the most strongly affected Khc resonances typically decreased by ~50%. This observation indicates an effect of Tm1 on half the Khc tail molecules. Considering the 2:1 stoichiometry of Khc–Tm1 (Supplementary Fig. 1)[32] we envisage Tm1 affecting only one chain of the Khc tail dimer. Because signals from the affected Khc tails were broadened beyond detection, we could not assess secondary chemical shifts and thus whether the corresponding tail region undergoes a conformational change in the Tm1-bound state. Nonetheless, the intensity changes observed for the IAK motif and its proximal residues indicate either an induced conformational change or a second interaction of the Tm1 N-terminal domain with this region. Therefore, although the IAK motif is not required for Tm1-mediated inhibition of Khc (Fig. 2a–c and Supplementary Table 1), its chemical environment is affected by Tm1 binding. These findings suggest that Tm1 modulates interaction of the Khc motor with the tail.

### Tm1 does not abrogate the Khc motor-tail interaction

To evaluate the effect of Tm1 on the Khc motor-tail interaction, we performed NMR experiments in which the motor domain (amino acids 1–365) was added to the tail domain (amino acids 855–975) in the presence and absence of Tm1 1–335. Without Tm1 1–335, titration of the motor domain to the tail caused intensity losses for several resonances within the tail, including those corresponding to the IAK motif (amino acids 941–948) and its flanking residues (amino acids 936–962; Fig. 4f,g). Some resonances in this region also showed chemical shift perturbations (Fig. 4f,h). These data are consistent with the previously characterized interaction of the IAK motif with the motor domain. Although most resonances of the Khc tail–motor complex were unaffected by addition of Tm1 1–335, those corresponding to the IAK motif and flanking residues saw further intensity decreases (Fig. 4f,g). Taken together, these data indicate that Tm1 can bind the Khc tail when it is complexed to the motor domain but does not abrogate the interaction of the isolated motor domain with the isolated tail domain. These conclusions were corroborated with glutathione S-transferase (GST)-tag pulldown experiments in which Khc 855–975 was tethered to beads and incubated with the motor domain in the presence and absence of Tm1 (Extended Data Fig. 4c,d).

**Fig. 4 | Tm1 binding affects the free and motor-bound Khc tail. a**, $^1$H-$^{15}$N HSQC spectrum of the Khc tail construct Khc 855–975 with backbone assignments indicated. **b,c**, $^1$H-$^{15}$N HSQC titration experiment with U10 RNA and Tm1 1–335. Overlaid spectra (**b**) and plotted intensity ratios of individual residues (**c**). RNA was added before Tm1. **d**, Intensity ratios of a titration experiment with Tm1 1–335 and U10 RNA in which Tm1 was added before RNA. **e**, Intensity ratios of a titration experiment with U10 RNA and Tm1 252–334 in which RNA was added

before Tm1. **f–h**, $^1$H-$^{15}$N HSQC titration experiment of the Khc tail (Khc 855–975) with the motor domain (Khc 1–365) and Tm1 1–335. Overlaid spectra (**f**), plotted intensity ratios (**g**) and chemical shift perturbation (CSP) plot (**h**) of individual residues are presented. Khc 1–365 was added before Tm1 1–335. Data are presented as measured value ± s.d. Error bars represent estimates of propagated measurement errors of the experimental uncertainties in signal amplitudes.

## Tm1 alters the autoinhibited conformation of Khc

We next reasoned that Tm1 binding might stabilize the autoinhibited conformation of Khc FL by favoring additional interactions within the coiled-coil stalk domain, likely including the AMB domain. To obtain information about Khc's predominant conformation in its free and Tm1-bound states, we measured small-angle X-ray scattering (SAXS) of

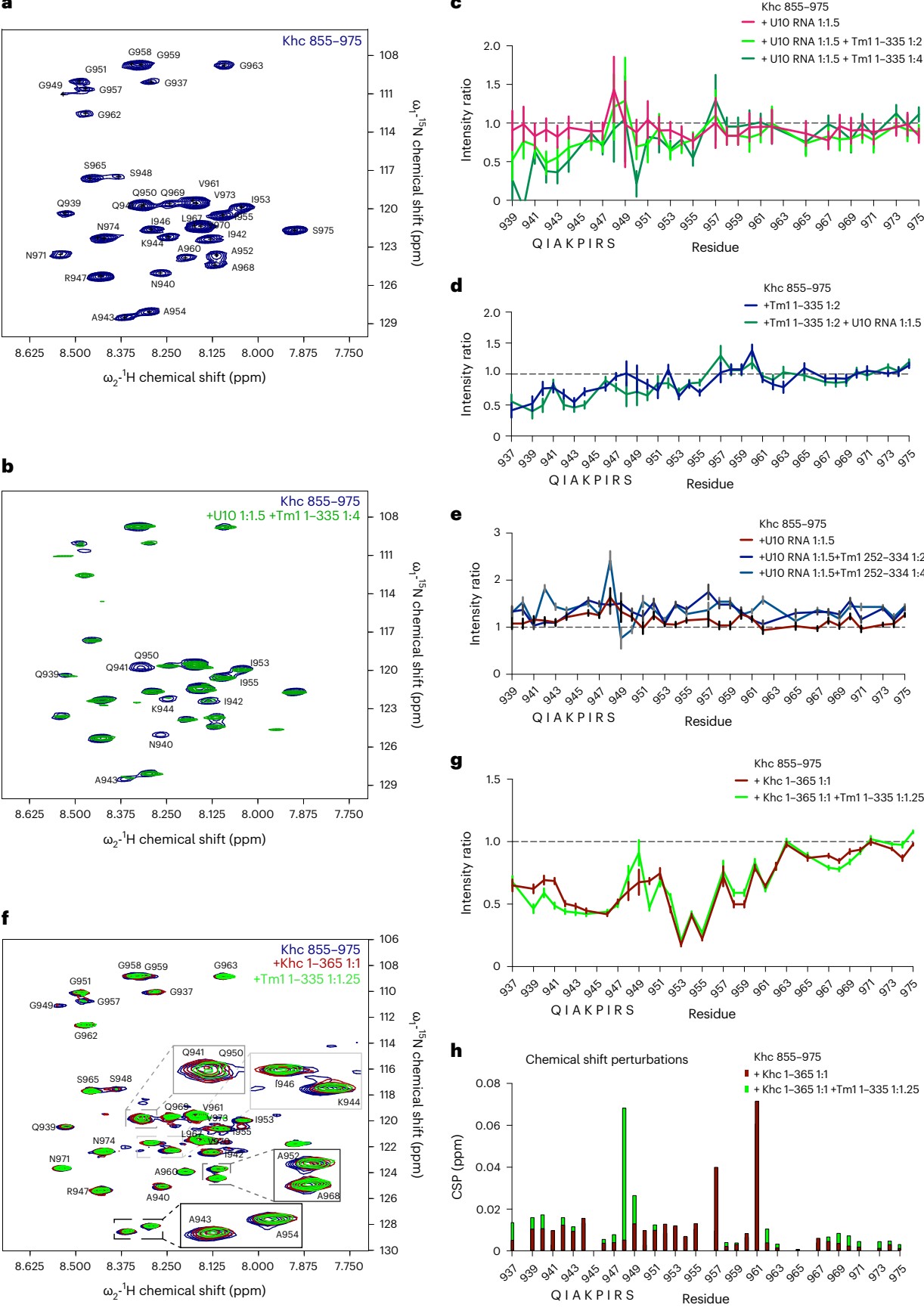

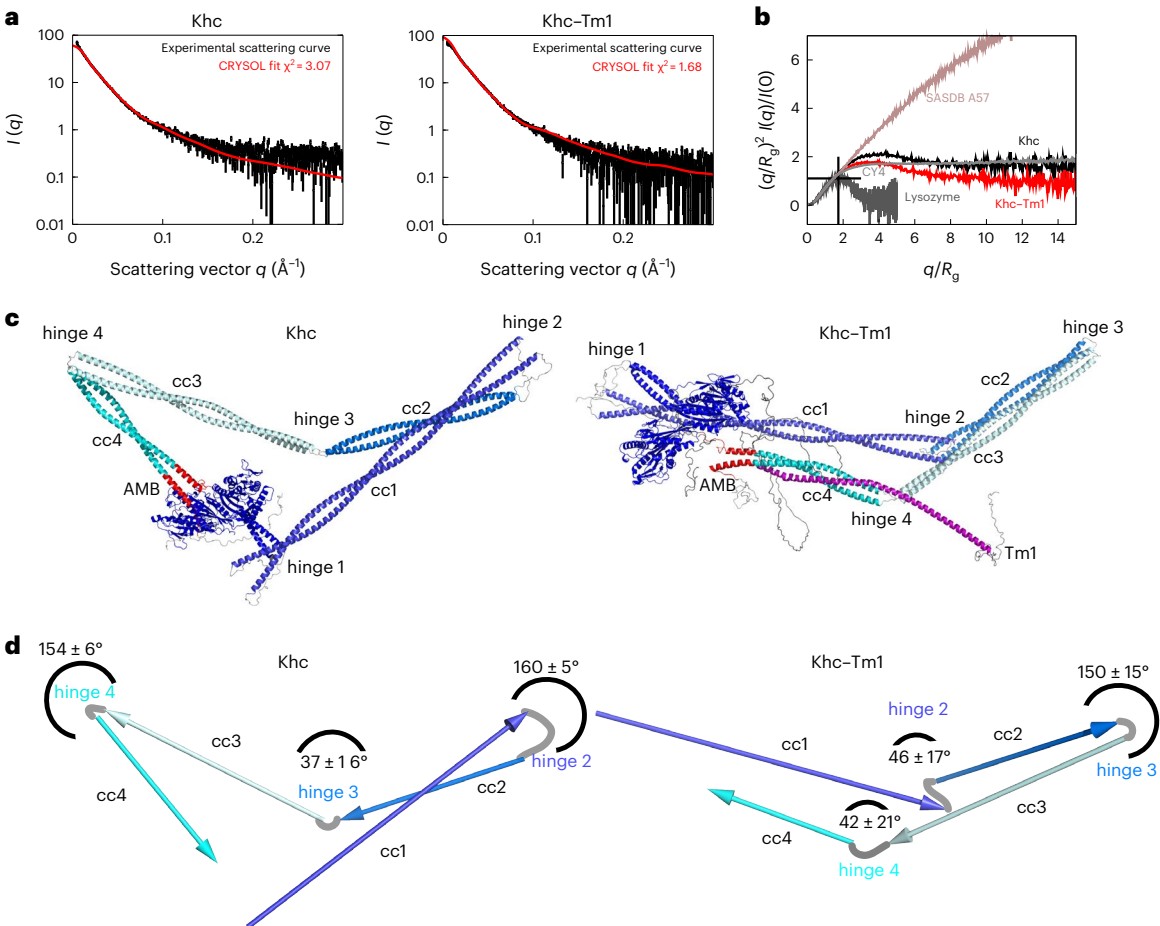

**Fig. 5 | Tm1 binding promotes compaction of the folded Khc conformation.**
**a**, Experimental SAXS curves overlaid with theoretical SAXS curves of best CRYSOL fits ($R_g$) for Khc (left) and Khc–Tm1 (right). $I(q)$ is the scattering intensity at scattering vector $q$. **b**, Dimensionless Kratky plot of scattering profiles of Khc and the Khc–Tm1 complex compared with a globular protein (lysozyme), a disordered protein with short secondary structure elements (SASBDB[68] CY4) and a fully disordered extended protein (SASBDB[68] A57) to illustrate differences between Khc and Khc–Tm1 and contributions of ordered and disordered regions. **c**, Best fits for Khc and Khc–Tm1. Khc models are shown in shades of blue with AMB domains marked in red; Tm1 is shown in pink. cc1–4 denote the four coiled-coil regions of the Khc stalk domain. **d**, Schemes showing the orientations of Khc's stalk coiled coils in the structures with best CRYSOL fits. Angles between coiled coils are means ± s.d. from the five best-fitting models.

Khc FL in isolation and in complex with Tm1 FL (Fig. 5a). A dimensionless Kratky plot[43,44] of both scattering profiles (Fig. 5b) showed that Khc and Khc–Tm1 are extended molecules with flexible elements. To identify structures that can explain the scattering curves, we generated large pools of Khc and Khc–Tm1 models by connecting coordinates of the structured parts generated by AlphaFold2 (refs. 45,46). Based on AlphaFold2 predictions, the Khc stalk contains four coiled-coil regions (cc1–4) connected by flexible hinge regions (hinges 2–4). The stalk is connected to the motor domain's neck coiled coil by another flexible region (hinge 1; Extended Data Fig. 5a). We subsequently randomized backbone angles in the connecting linker and tail regions of Khc to create a large pool of 5,000 possible structures for both Khc and Khc–Tm1. In a second step, the C-terminal tail of one Khc chain was pulled into the binding pocket of the motor domain resulting in a second pool of 5,000 structures for both Khc and Khc–Tm1 with random orientation of the stalk region and the IAK peptide bound to the motor domain dimer. Each pool of structures was sorted by the quality of the fit to the respective SAXS data.

For both Khc and Khc–Tm1, the structures with the lowest $\chi^2$ values (the best quality fits) all exhibited compact conformations with considerably lower than average radius of gyration ($R_g$) values. Because the autoinhibited conformations were more compact than the initial pool with randomized linker conformations, there were more structures with low $\chi^2$ values in the pools of structures with the IAK motif bound to the motor domains (Extended Data Fig. 5b). Inspection of the structural models that best fit the scattering curves indicated that the arrangement of the Khc coiled coils is different for Khc and Khc–Tm1 (Extended Data Fig. 5c). In the ensemble of well-fitting Khc models (Extended Data Fig. 5d), the conformation of the Khc hinge regions (hinge 2: amino acids 586–603; hinge 3: amino acids 707–710; and hinge 4: amino acids 839–843) (Extended Data Fig. 5a) leads to bent structures of the coiled-coil regions cc1–cc2 and cc3–cc4, as well as an elongated cc2–cc3 orientation, resulting in a triangular shape of the molecule (Fig. 5c,d and Extended Data Fig. 5d). In this conformation, the N- and C-terminal parts of the Khc stalk are separated, and the AMB domain, which mediates inhibition by Tm1, is not in proximity with other stalk regions (Fig. 5c,d and Extended Data Fig. 5d,e). In well-fitting Khc–Tm1 models, the stalk folds at hinge 3 (Fig. 5c,d and Extended Data Fig. 5d), bringing the N- and C-terminal halves of the stalk in closer proximity than observed for Khc alone, with the AMB domain oriented parallel to the N-terminal stalk region (Extended Data Fig. 5f).

During the course of our work, two studies[47,48] proposed compact structural arrangements for mammalian kinesin-1 with a more rigid stalk that cannot fold at hinge 2. We therefore repeated our modeling with new structure pools resembling this rigid stalk conformation. The resulting structure that best fits the SAXS data for Khc exhibits a

$x^2$ increase of 15% compared with our previous best-fitting model (3.54 compared with 3.07; Extended Data Fig. 6a,b), suggesting a less-good fit. In all of the best-fitting models, the stalk exhibits a larger distance between the cc3–cc4 linker region and cc1–cc2 and thus, similar to our original models, a triangular stalk arrangement, rather than a compact packing of cc3 and cc4 onto cc1–cc2 (Extended Data Fig. 6c). By contrast, the best-fitting Khc–Tm1 models have a topology and $x^2$ values comparable with those of the structures obtained by our original models (Extended Data Fig. 6a,b), which showed a similar compact arrangement with folding of the stalk at hinge 3 (Extended Data Fig. 6c). Thus, we find similar results using both flexible and rigid stalk arrangements. Although compact stalk arrangements do not fit our SAXS data for Khc, they fit well for the Khc–Tm1 complex. Although our fits cannot account for alternative Khc conformations (both autoinhibited and open) that may exist as part of dynamic conformational equilibria, the good agreement of individual models with SAXS data suggests that they represent predominant conformations for Khc with and without Tm1. These data therefore suggest that binding of Tm1 to Khc results in a shift in the conformational equilibrium towards more compact structures of the Khc stalk domain, and that the autoinhibited conformation of Khc is likely to be stabilized by interactions within the Tm1-bound stalk in addition to the motor–tail interaction. This is in agreement with our finding that the Khc AMB domain can mediate inhibition of motor activity by Tm1 independently of the IAK motif.

## Tm1 links inhibited Khc to dynein-driven *osk* RNPs

As described in the Introduction, Khc and Tm1 are loaded onto *osk* RNPs in nurse cells before their delivery into the oocyte by dynein[31]. We reasoned that during this process enhanced inhibition of Khc by Tm1 would prevent opposition to dynein motility that could result in counterproductive movements of RNPs away from the oocyte. To assess whether this is the case, we set out to reconstitute microtubule-based transport of *osk* RNPs containing Khc FL, Tm1 FL and dynein.

Because dynein and kinesin require different conditions for optimal motility in vitro, we used a modified assay buffer that allows movement of both motor classes[49]. To exclude the possibility that Tm1 promotes minus-end directed movement of *osk* RNPs by stimulating dynein directly, we first assessed its effects on the motile properties of dynein in the presence of DDBE and an *osk* 3′ UTR fragment that is sufficient for dynein-mediated transport in vivo and in vitro (*osk* 3′ UTR 2+3)[34,37]. Although Tm1 could translocate with dynein and *osk* (Extended Data Fig. 7a), its presence did not alter dynein velocity or run length (Extended Data Fig. 7b,c and Supplementary Table 3). Furthermore, Tm1 did not affect dynein binding to microtubules or the frequency of processive motion (Extended Data Fig. 7d–f and Supplementary

Table 3). Thus, Tm1 does not directly influence dynein motility. Treatment of these reconstituted RNPs with RNase A dramatically reduced association of Tm1 with DDBE–*osk* 3′ UTR 2+3 (Extended Data Fig. 7g,h), indicating that Tm1 is linked to the dynein machinery through *osk* RNA rather than protein–protein interactions.

We next assessed the behavior of Khc FL in the presence of DDBE and *osk* 3′ UTR 2+3 but without Tm1. In these conditions, dynein and associated *osk* RNA exhibited many long movements in the minus-end direction (Fig. 6a), as shown by polarity marking of the microtubules (Supplementary Fig. 5). By contrast, movement of Khc FL on microtubules was relatively infrequent (Fig. 6a,b and Supplementary Table 4). Plus-end directed runs accounted for only ~22% of observed Khc motility (Extended Data Fig. 8a), and were seldom associated with *osk* (Fig. 6a,c). The remainder of these infrequent Khc movements were short and towards the microtubule minus ends in transient association with dynein and the RNA (Fig. 6a,c and Extended Data Fig. 8a), revealing some inherent capacity of Khc FL to associate with DDBE–*osk* RNA complexes. Khc motors in these complexes appeared to retain some activity, as they occasionally exhibited sudden plus-end directed movements following periods of minus-end directed transport by dynein (9 of 127 complexes (7.1%); Extended Data Fig. 9).

Inclusion of Tm1 FL in the assay caused an approximately tenfold increase in the incidence of Khc transport events (Fig. 6a,b and Supplementary Table 4), almost all of which were minus-end directed (Extended Data Fig. 8a) and overlapped with dynein and *osk* RNA signals (Fig. 6c). Although plus-end directed Khc FL movements were very rare in the presence of Tm1, they were more likely to colocalize with dynein and *osk* RNA (45% of complexes) than were plus-end directed movements of the motor in the absence of Tm1 (13% of complexes; $P = 0.003$, Fisher's exact test) (Fig. 6c). Collectively, these findings demonstrate that Tm1 enhances association of Khc with DDBE–*osk*, and that this imparts a very strong minus-end directed bias to Khc movement. Treatment with RNase A strongly suppressed the Tm1-induced association of Khc with DDBE–*osk* (Extended Data Fig. 10), revealing that Tm1 achieves this effect via binding to the RNA.

Tm1 also increased the distance Khc FL traveled in association with minus-end directed DDBE–*osk* RNPs by ~60% (Fig. 6d and Supplementary Table 4) and reduced the incidence of plus-end directed Khc FL movements following minus-end directed DDBE–*osk* motion (6 of 848 complexes (0.7%); $P < 0.0001$ compared with in the absence of Tm1 (Fisher's exact test)). Together, these data are consistent with Khc activity in these complexes being curbed by Tm1. To explore this idea, we compared velocities of Khc-bound dynein complexes in the presence and absence of Tm1. Consistent with an inhibitory effect of Tm1 on Khc, DDBE–*osk* RNA complexes bound by Khc moved towards the minus end

**Fig. 6 | Tm1 promotes association of inhibited Khc with minus-end directed DDBE–*osk* RNPs. a**, Kymographs showing motile behavior of DDBE (via tetramethylrhodamine (TMR)-dynein), Alexa Fluor 647 (A647)-Khc FL and Alexa Fluor 488 (A488)-*osk* 3′ UTR 2 + 3 in the presence or absence of unlabeled Tm1 FL. Plus- and minus-end directed Khc movements in the absence of Tm1 FL are indicated by white and yellow arrowheads, respectively. Microtubule plus and minus ends are oriented toward the right and left of each kymograph, respectively. **b**, Frequency of processive Khc FL and Khc940 movements towards microtubule minus ends (towards left of *x* axis, purple) and plus ends (towards right of *x* axis, magenta) with or without Tm1 FL. Mean ± s.d. is shown and is derived from 60–88 microtubules (*N*), representing analysis of 178–1,795 individual complexes from 5–6 imaging chambers and 3 independent experiments (*n*) per condition (Supplementary Table 4). Mean values for each independent experiment (large dots) are superimposed on values for individual microtubules (small dots). **c**, Cumulative fraction of plus- and minus-end directed Khc FL and Khc940 signals that comigrated with other labeled assay components with or without Tm1 FL. Rare instances of minus-end directed Khc FL that do not colocalize with dynein signal are likely due to incomplete dynein labeling or photobleaching. Numbers refer to total number of complexes (*N*)

analyzed and independent experiments for each condition (*n*) in the format *N(n)*. **d**, Distance traveled by Khc FL towards minus end in the presence or absence of Tm1 FL. 1-Cumulative frequency distributions of measured run lengths were fit as one-phase decays with the decay constant τ ± 95% confidence interval shown. Hollow triangles represent empirical bin values used for fitting and were derived from analysis of 158–1,259 individual complexes (*N*) from 3 independent experiments (*n*) per condition (Supplementary Table 4). **e**, Velocity of minus-end directed DDBE–*osk* 3′ UTR 2 + 3 complexes with comigrating Khc FL signal in the presence or absence of Tm1 FL. Median ± interquartile range is shown and is derived from analysis of 158–1,259 individual complexes (*N*) from 5–6 imaging chambers and 3 independent experiments (*n*) per condition (Supplementary Table 4). Median values for each independent experiment (large dots) are superimposed on a violin plot showing distribution of values from individual complexes. Black or white circles indicate the presence or absence of the indicated components, respectively. In **b** and **e**, parallel experiments are shown with large dots of the same color. Statistical significance was determined by an unpaired two-tailed *t*-test and *t*-test with Welch's correction (**b**) or unpaired two-tailed Mann–Whitney test (**d,e**) using *N* values (total number of microtubules (**b**) or total number of individual complexes (**d,e**)). NS, not significant (*P* > 0.05).

nearly twice as fast in the presence of Tm1 (Fig. 6e and Supplementary Table 4). This observation indicates that Tm1 reinforces the autoinhibition conferred by the Khc tail, presumably by limiting engagements of Khc with microtubules that oppose dynein movement. Our data support a model in which Tm1 enables efficient minus-end directed transport of Khc and dynein-associated RNPs from the nurse cells into the oocyte by coupling Khc to these structures in a strongly inhibited state.

To determine whether Tm1 can also promote minus-end directed transport of a Khc motor by DDBE–*osk* when Khc autoinhibition by the IAK motif is relieved, we performed equivalent experiments with the Tm1-sensitive Khc940 tail truncation. In the absence of Tm1, we

observed a large number of processive Khc940 movements (Fig. 6b and Extended Data Fig. 8b), the majority of which (73%) were directed towards microtubule plus ends (Extended Data Fig. 8a,b) independently of dynein or RNA (Fig. 6c). This observation is in keeping with the activation of motility that we (Fig. 2) and others[9–12] have seen with isolated Khc lacking the IAK motif. The infrequent minus-end directed movements of Khc940 were almost always associated with dynein and *osk* signals (Fig. 6c), consistent with our observation of some interaction of Khc FL with dynein in the absence of Tm1.

The presence of Tm1 was sufficient to impart an overall minus-end bias to Khc940 transport events, with only 42% of these motors now

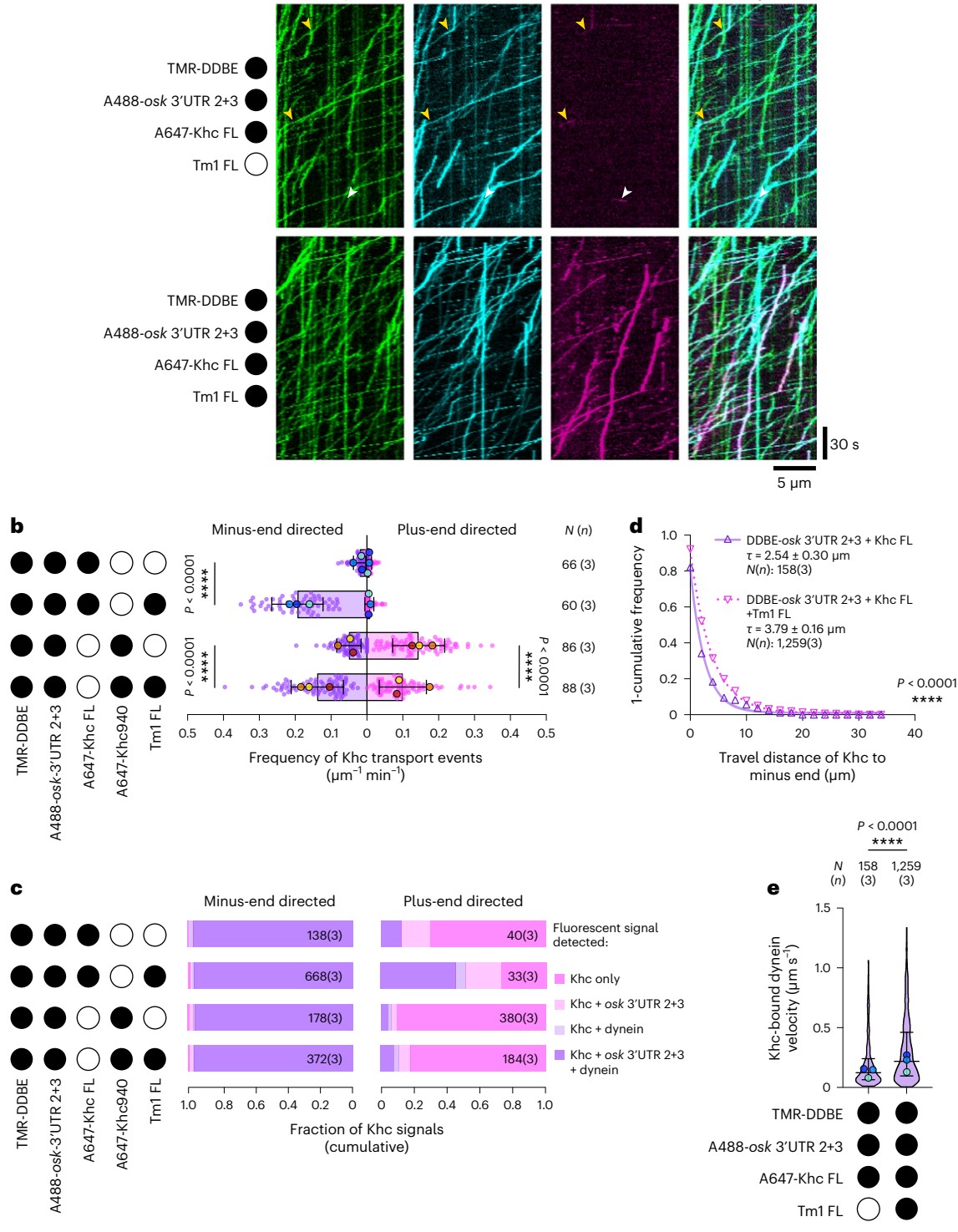

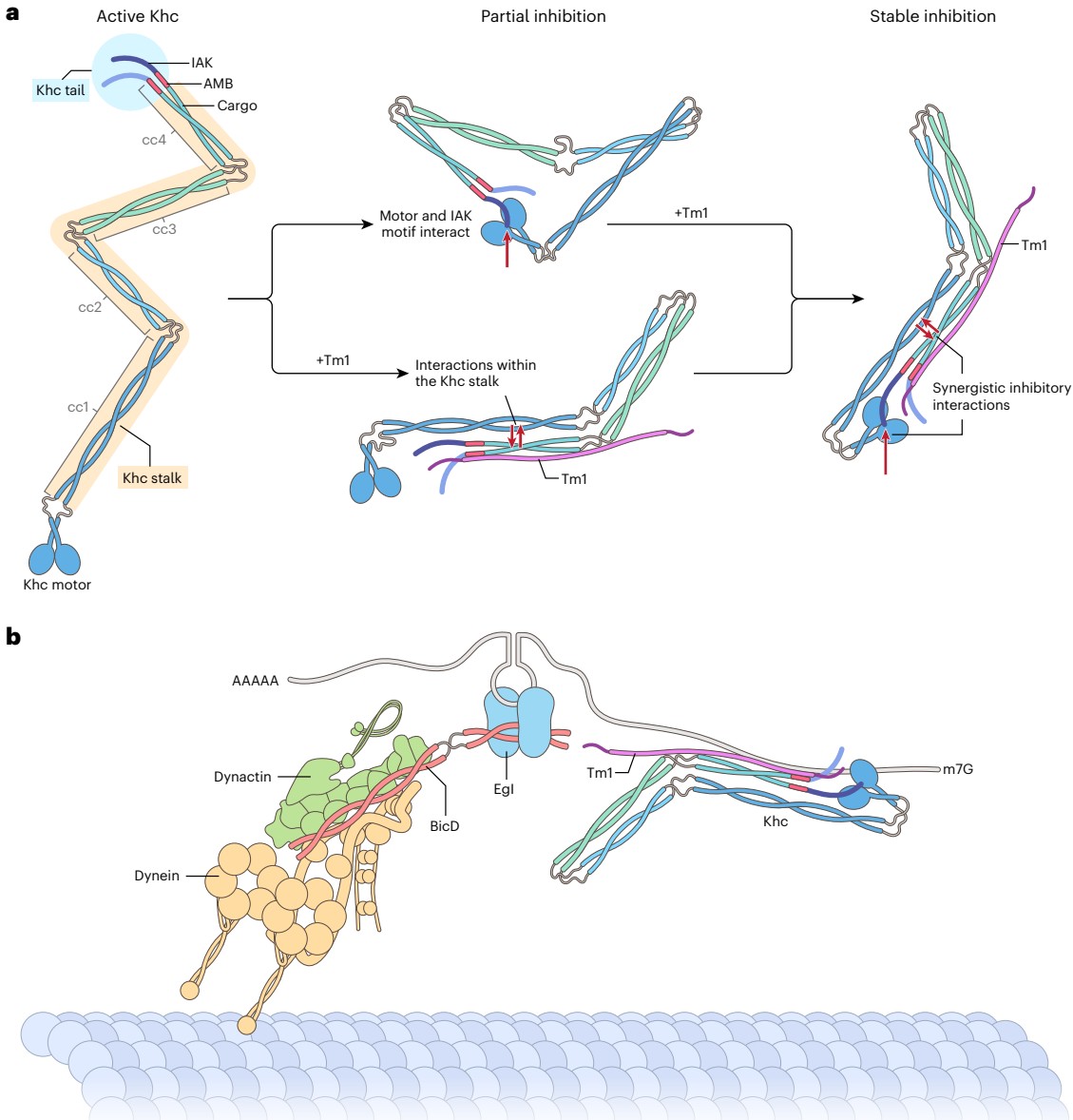

**Fig. 7 | Model of Khc regulation by Tm1 in *osk* RNPs. a**, Model of Tm1-mediated stabilization of autoinhibited conformation of Khc. Whereas both the motor domain-IAK motif interaction and interactions within the Tm1-bound, rearranged Khc stalk region (including the AMB domain) can partially inhibit Khc, synergy between these weak interactions stabilizes the inhibitory head-tail interaction. **b**, Model of Tm1's role in dynein-mediated transport of *osk*. By linking Khc to the dynein-transported *osk* RNA and promoting Khc inhibition, Tm1 allows the first, dynein-mediated transport step from the nurse cells to the oocyte to proceed without competition between the motors. These events also position Khc on the RNP for the second, Khc-mediated transport step within the oocyte. AAAAA and m7G indicate poly A tail and cap structure of *osk*, respectively.

plus-end directed (Extended Data Fig. 8a,b). This was associated with an approximately threefold increase in the frequency of minus-end directed motion in association with DDBE–*osk* at the expense of plus-end directed runs (Fig. 6b). Unlike the case with Khc FL, minus-end directed dynein complexes bound to Khc940 did not move significantly faster or for greater distances in the presence of Tm1 (Extended Data Fig. 8c,d). This may be because the truncated motor is more competent to bind microtubules than Khc FL even when Tm1 is present (Extended Data Fig. 1) and can therefore resist dynein motion regardless of Tm1 association. Nonetheless, these observations show that Tm1 is sufficient to impart a minus-end directed bias to the movement of Khc motors in the presence of DDBE–*osk* even without the inhibitory effect of the IAK motif. Thus, shutting down Khc activity during minus-end directed *osk* RNP transport requires multiple inhibitory interactions involving the Khc tail.

## Discussion

Tm1 was recently implicated as an RNA adaptor for kinesin-1 (refs. 30–32). Our current study reveals a previously unknown role of Tm1 in *osk* transport. We show that Tm1 negatively regulates Khc activity, which we propose occurs via a conformational change in the Khc stalk that stabilizes the Khc motor-tail interaction and thereby enhances autoinhibition (Fig. 7a).

With its functions in cargo binding[32] and motor regulation, we speculate that Tm1 is an alternative Klc in the *Drosophila* female germline[29]. Because precise *osk* RNA localization during oogenesis is critical for development, positive regulators of Khc function such as PAT1[28] and negative regulators such as Tm1 may have replaced Klc to provide nuanced control of Khc activity. If our hypothesis is correct, Klc should also be dispensable for Tm1-dependent RNA localization in somatic tissues[50,51].

We find that Tm1 also stimulates association of Khc in a strongly inhibited state with dynein-associated *osk* RNPs. This mechanism would allow dynein-mediated transport of *osk* RNPs from the nurse cells to the oocyte to proceed without competition with Khc, while positioning the plus-end directed motor on the RNPs for their posteriorward transport within the ooplasm in mid-oogenesis (Fig. 7b).

### Kinesin-1 autoinhibition mechanisms

Kinesin-1 autoinhibition is not fully understood, in part because of a paucity of structural information for the full-length molecule. It has been proposed that interaction of the tail's IAK motif with the motor domain plays a key role in kinesin-1 autoinhibition. Consistent with this notion, we observed strongly enhanced motility of *Drosophila* Khc when the IAK motif was deleted. However, it was recently shown that Klc inhibits Khc activity independently of the IAK-motor interaction[14]. We also find that the IAK motif is not needed for inhibition of Khc by Tm1. Instead, we find that Tm1-mediated inhibition occurs via the Khc AMB domain, a region adjacent to the IAK motif that is essential for *osk* localization[52].

Our structural analyses suggest that Tm1 stabilizes the autoinhibited folded conformation of Khc by inducing rearrangement of the Khc coiled-coil stalk. We therefore propose a model in which both the motor-IAK interaction and interactions within the Tm1-bound stalk that include the AMB domain act synergistically to achieve the stable inhibited conformation of Khc (Fig. 7a).

Two recent studies have proposed compact structural arrangements for mammalian kinesin-1 (refs. 47,48). Those studies used chemical crosslinking and cryo-EM, which are likely to enrich for a homogeneous population of compact conformations of Khc and Khc–Klc tetramers, to provide detailed static structural information. By contrast, in our study of *Drosophila* Khc and Khc–Tm1 complexes, we employed NMR and SAXS, which provide insight into the different conformational states, and thus flexibility, of kinesin-1 by enabling analysis of structures in solution. Tan et al. observed the compact, inhibited conformation in the isolated Khc, showing that Klc-binding does not induce a new fold of Khc but rather stabilizes the inhibited conformation[47]. This is in agreement with our model, in which Tm1, a putative alternative Klc, shifts the structural equilibrium of Khc towards the autoinhibited state by stabilizing its compact conformation.

### Coordination of kinesin-1 and dynein on *osk* RNPs

Although it is known that many cargo types are transported by the concerted action of dynein and kinesins, the underlying regulatory mechanisms have been elusive. We have identified one of very few examples of a factor that not only links dynein and kinesin-mediated transport, but also modulates transport through differential motor regulation. Linkage of dynein and kinesin-3 by the dynein-activating adaptor Hook3 has been demonstrated[53,54], but the cellular events for which this is relevant are still emerging. Recently, reconstituted coupling of dynein and kinesin-1 by TRAK1 and TRAK2 has provided insight into how the motors are recruited and regulated for mitochondrial transport[55,56]. However, unlike our integration of DDBE and Khc in reconstituted *osk* RNPs, these systems lacked intact native cargoes and thereby excluded potential cargo-directed positioning and modulation of motor complexes.

Several studies have reported that active dynein and kinesin motors engage in a tug-of-war when artificially coupled[57–62]. We also observe motor opposition in reconstituted RNPs containing DDBE and Khc, presumably because of the stochastic engagement of autoinhibited Khc with microtubules. However, our observation that Tm1 supports efficient dynein-mediated RNA transport through robust inhibition of Khc highlights the importance of regulatory factors in addition to mechanical coupling in native transport complexes. Supporting the in vivo relevance of negative regulation of Khc during bidirectional transport, kinesin-1-activating IAK mutations were recently shown to impair dynein-mediated transport processes in *Aspergillus nidulans*[63]. Collectively, these observations point to complex interplay between opposite-polarity motors that are bound simultaneously to cargoes. Further reconstitutions of dual motor systems on native cargoes should reveal generalities of dynein-kinesin crosstalk, as well as any cargo-specific regulatory mechanisms.

### Kinesin-1 activation mechanisms in the female germline

Our study provides mechanistic insight into two critical aspects of *osk* mRNA transport—assembly of the dual motor complex and how Khc activity is suppressed during dynein-mediated delivery of the transcript from the nurse cells to the oocyte. However, it is not understood how Khc takes over from dynein after *osk* RNPs arrive in the oocyte. Although our recent work has shown that inactivation of dynein by the RNA-binding protein Stau is part of this process[34], how Tm1- and IAK-mediated inhibition of Khc is alleviated to allow delivery of the mRNA to the oocyte posterior is an open question. One candidate to fulfill this role is Ensconsin, which is required for posterior *osk* localization and is enriched in the oocyte relative to the nurse cells[64]. Strikingly, the human counterpart of Ensconsin (MAP7) was recently shown to stimulate activity of mammalian kinesin-1 in vitro[14,65,66]. Other candidate Khc activators include the exon junction complex, which, together with the SOLE RNA structure, is essential for transport of *osk* to the oocyte posterior[20,35,67]. Because Tm1 needs to remain bound to the *osk* RNP throughout its posterior translocation[31,32], it is likely that the activating factor(s) induces a conformational change in the Khc–Tm1 complex rather than dissociation of Tm1. Future investigations of these regulatory mechanisms are likely to elucidate how kinesin-1 activity is orchestrated in other systems.

## Online content

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

## Methods

### Molecular cloning

Khc FL for recombinant expression was cloned with a HisSUMO-SNAP-3C-tag into multiple cloning site 1 of the pFastBacDual vector by InFusion cloning (Takara) (between BamHI and HindIII sites). The Khc mutant constructs were generated by site-directed mutagenesis of the Khc FL plasmid using primers that either inserted a stop codon after amino acid 910 (Khc910) or amino acid 940 (Khc940), or deleted amino acids 911–938 (KhcΔAMB), followed by digestion of the template plasmid with Dpn1 and transformation of the product into *Escherichia coli* XL1-blue cells. For coexpression of Khc FL and Tm1 FL (used for mass photometry and SAXS experiments), the Tm1 coding sequence was inserted into multiple cloning site 2 of pFastBacDual-HisSUMO-SNAP-3C-Khc by InFusion cloning (between SmaI and KpnI sites). Tm1 FL for in vitro motility assays was cloned with a HisSUMO-SNAP-tag between the SacI and HindIII sites of the pET11 vector by InFusion cloning. The SNAP-Tm1 1–335 truncation was made by site-directed mutagenesis of the SNAP-Tm1 FL plasmid using primers that inserted a stop codon after amino acid 335, as described above. Khc 855–975 and Khc 1–365 were cloned into pGEX6.1 by InFusion cloning (between BamHI and EcoRI sites).

### Protein expression and purification

N-terminally His6-SUMO-SNAP-3C-tagged Khc FL, Khc910 or 940 and ΔAMB truncations, and coexpressed Khc–Tm1 complexes were expressed using the Bac-to-Bac Baculovirus expression system (Gibco). One liter of *Sf21* insect cells at $1 \times 10^6$ cells ml$^{-1}$ of shaking culture were infected with 10 ml of P1 virus stock. After expression for 3 days at 27.5 °C, cells were harvested by centrifugation at 1,000g for 20 min at 4 °C. The pellets were lysed in a dounce tissue grinder in lysis buffer (20 mM Tris–HCl at pH 7.5, 500 mM NaCl, 1 mM MgCl$_2$, 0.1 mM ATP, 2 mM DTT, 5% glycerol, 0.2% Tween-20). The lysates were cleared by centrifugation and the soluble protein fraction was affinity-purified on a HisTrap Excel column (GE Healthcare). After elution in a 0–300 mM imidazole gradient, the HisSUMO-fusion tag was cleaved either by Senp2 protease (SNAP-tagged proteins for fluorescent labeling) or 3C protease (unlabeled proteins) during dialysis in GF150 buffer (25 mM HEPES/KOH pH 7.3, 150 mM KCl, 1 mM MgCl$_2$, 0.1 mM ATP, and 2 mM DTT) for 16 h before further purification by anion-exchange chromatography on a HiTrap Q HP column (GE Healthcare), followed by size-exclusion chromatography (SEC) on a Superdex 200 Increase 10/300 column (GE Healthcare).

His6-SUMO-tagged and His6-SUMO-SNAP-tagged Tm1 proteins and GST-Khc 1–365 were expressed in BL21-CodonPlus(DE3)-RIL cells (Stratagene) by IPTG induction for 16 h at 18 °C. Cells were grown in Luria–Bertani medium (0.5% w/v yeast extract (MP Biomedicals), 1% w/v tryptone (MP Biomedicals), 0.5% w/v NaCl at pH 7.2) supplemented with antibiotics (100 µg ml$^{-1}$ ampicillin or 10 µg ml$^{-1}$ kanamycin and 34 µg ml$^{-1}$ chloramphenicol). After harvesting, the pellets were lysed using a microfluidizer processor (Microfluidics) in 20 mM Tris–HCl (pH 7.5), 500 mM NaCl, 5% glycerol, 0.01% NP-40 and 40 mM imidazole buffer supplemented with protease inhibitor cocktail (Roche) and 2 mM DTT. The lysates were subsequently cleared by centrifugation. HisSUMO-tagged proteins were affinity-purified by Ni-IMAC on a HisTrap column (Cytiva) and eluted with an imidazole gradient (40–300 mM). After cleavage of the His6-SUMO-tag with Senp2 protease, cleaved tags and protease were removed by a second, subtractive Ni-IMAC on a HisTrap column (GE Healthcare). The proteins were further purified on either a Superose 6 Increase 10/300 GL or a HiLoad 16/600 Superdex 200 SEC column in either GF150 buffer for motility assays or NMR buffer (25 mM NaPO$_4$ pH 6.5, 150 mM NaCl and 0.5 mM Tris(2-carboxyethyl)phosphin for NMR experiments.

GST-tagged Khc 1–365 was affinity-purified on a GSTrap column (Cytiva) and glutathione gradient (1–20 mM). For use in NMR experiments, the GST tag was removed by cleavage with 3C protease and a second, subtractive affinity purification step on a GSTrap column. The protein was further purified on a HiLoad 16/600 Superdex 200 SEC column in GF150 buffer or NMR buffer.

SNAP-tagged Khc and Tm1 proteins were fluorescently labeled with SNAP-Cell TMR Star, SNAP-Surface Alexa Fluor 488 or SNAP-Surface Alexa Fluor 647 at 15 °C for 90 min in GF150 buffer with a 1.5× molar excess of dye. Free dye was removed by a subsequent desalting step on a PD-10 desalting column (Cytiva). Labeling efficiencies were determined by measuring protein concentration at 280 nm and dye concentration at the dye's excitation maximum using a NanoDrop 8000 Spectrophotometer (Thermo Fisher Scientific). For monomeric protein, labeling efficiency was calculated as $c_{dye}/c_{protein}$. For dimeric proteins, labeling efficiency was calculated based on monomer labeling efficiencies as $(L_1 \times U_2) + (U_1 \times L_2) + (L_1 \times L_2)$ where $L_1$ is the fraction of monomer 1 labeled, $L_2$ is the fraction of monomer 2 labeled, $U_1$ is the fraction of monomer 1 unlabeled and $U_2$ is the fraction of monomer 2 unlabeled.

Recombinant human dynein and *Drosophila* Egl-BicD complexes were expressed in *Sf9* insect cells with the dynein complex fluorescently labeled via a SNAPf tag fused to the N terminus of the dynein heavy chain, as described previously[25]. Native porcine dynactin was purified from brain tissue as described previously[25].

### RNA in vitro transcription

The *osk* RNA constructs used for the motility assays were synthesized in vitro using the MEGAscript T7 Transcription Kit (Ambion). The RNA was transcribed from a PCR product amplified with a forward primer including the T7 promoter and containing an *osk* mRNA FL, *osk* 3′ UTR or a 529-nucleotide region of the *osk* 3′ UTR previously defined as region 2+3 (*osk* 3′ UTR 2+3) that is sufficient to promote the localization of reporter transcripts to the developing oocyte[37]. The template for *osk* min was generated from three overlapping primers by PCR and subsequently amplified again with a T7 promoter-containing forward primer. RNA was transcribed from the final PCR product. For fluorescent labeling of the *osk* 3′ UTR and *osk* min transcripts, cyanine3 (Cy3)- or Cy5-uridine-5'-triphosphate (UTP) was included in the transcription reaction with a fourfold excess of unlabeled UTP, yielding transcripts that typically had several labels per RNA molecule (close to 100% of molecules containing at least one dye). For *osk* 3′ UTR 2+3, transcripts were fluorescently labeled with Alexa Fluor 488-UTP with a fivefold excess of unlabeled UTP, which also resulted in close to 100% labeling efficiency. Transcripts were subsequently purified by phenol/chloroform extraction and passed through a G50 mini Quick Spin RNA desalting column (Sigma-Aldrich) to remove unincorporated nucleotides before precipitation with sodium acetate or ammonium acetate plus ethanol. The precipitated RNA was resuspended in nuclease-free water and stored at −20 °C or −70 °C.

### Polymerization and stabilization of microtubules

Microtubules were polymerized from porcine tubulin (Cytoskeleton) and labeled with fluorophores and biotin by stochastic incorporation of labeled dimers into the microtubule lattice. Mixes of 2.5 µM unlabeled tubulin, 0.17 µM HiLyte 488 tubulin or AMCA tubulin and 0.3 µM biotin-tubulin were incubated in BRB80 (80 mM PIPES pH 6.85, 2 mM MgCl$_2$, 0.5 mM EGTA, 1 mM DTT) with 0.67 mM GMPCPP (Jena Bioscience) overnight at 37 °C. Polymerized microtubules were pelleted in a room temperature table-top centrifuge at 18,400g for 8.5 min, and washed once with prewarmed (37 °C) BRB80. After pelleting once more, the microtubules were gently resuspended in 300 µl of prewarmed (37 °C) BRB80 containing 20–40 µM paclitaxel (taxol; Sigma-Aldrich).

Polarity marked microtubules with HiLyte 488 tubulin enriched at minus ends were made according to the principle described previously[69]. Bright microtubule seeds were polymerized at 37 °C for 10 min in a mix containing 2.5 µM unlabeled porcine tubulin (fivefold excess over fluorescent tubulin), 0.5 µM HiLyte 488 tubulin and 0.3 µM biotin-tubulin in BRB80 with 0.67 mM GMPCPP. After 10 min, this mixture was diluted 1:1 with a tubulin polymerization mix containing

12.5 µM unlabeled tubulin and 0.3 µM biotin-tubulin in BRB80 with 0.67 mM GMPCPP for final tubulin concentrations of 7.5 µM unlabeled tubulin (30-fold excess over fluorescent tubulin), 0.25 µM HiLyte 488 tubulin and 0.3 µM biotin-tubulin. The final tubulin mix was incubated at 37 °C for an additional 2 h to allow growth of relatively dim microtubule plus ends before harvesting and taxol stabilization as described above.

## Single-molecule total internal reflection fluorescence assays and image analysis

Total internal reflection fluorescence (TIRF)-based single-molecule reconstitution assays were performed as described previously[25]. Briefly, taxol-stabilized porcine microtubules were immobilized in imaging flow chambers by streptavidin-based linkages to biotin-PEG passivated cover slips. Khc RNP assay mixes were assembled at 500 nM of each component. Assemblies of the dynein transport machinery with Khc and Tm1 contained 100 nM dynein, 200 nM dynactin, 500 nM Egl–BicD, 1 µM RNA, 500 nM Khc (or Khc storage buffer as a control) and 2 µM Tm1 (or Tm1 storage buffer as a control), and were incubated on ice for 1–2 h before imaging. These mixes were then diluted to concentrations suitable for imaging of single molecules (6.25 nM for Khc RNP experiments; 5 nM dynein and 25 nM Khc for experiments containing both motors) in BRB12 (Khc experiments) or modified dynein motility buffer[49] (30 mM HEPES-KOH pH 7.0, 50 mM KCl, 5 mM MgSO$_4$, 1 mM EGTA pH 7.5, 1 mM DTT, 0.5 mg ml$^{-1}$ BSA—experiments including DDBE) supplemented with 1 mg ml$^{-1}$ α-casein, 20 µM taxol, 2.5 mM MgATP and an oxygen-scavenging system (1.25 µM glucose oxidase, 140 nM catalase, 71 mM 2-mercaptoethanol, 25 mM glucose final concentrations) to reduce photobleaching, and applied to the flow chamber for imaging by TIRF microscopy.

For assays in which RNase A was added, assemblies were diluted in modified dynein motility buffer as described above with either 1 µl of 0.2 µg ml$^{-1}$ RNase A (final concentration of 9.5 ng ml$^{-1}$) diluted in motility buffer, or 1 µl of motility buffer alone. Such low concentrations of RNase A were used to ensure mild digestion of the RNA (which was monitored during acquisition via the fluorescent RNA signal) that would eliminate any RNA-based connections between protein components while limiting digestion of the predominantly double-stranded localization signal important for activating DDBE motility[37]. These mixes were incubated at room temperature for 5 min before application to the imaging chambers.

For each chamber, acquisitions of 500 frames were made at a frame rate of ~2 frames s$^{-1}$ and 100 or 150 ms exposure (one or two colors) or 1.2–1.4 frames s$^{-1}$ and 50 ms exposure (three colors). Image series of Khc–Tm1 were acquired using a Leica GSD TIRF microscope system using Leica LAS X control software. All images involving dynein were acquired on a Nikon TIRF system using Micromanager control software[70], as described previously[25].

Binding and motility of RNP components in the reconstitution assays were manually analyzed by kymograph in Fiji[71] as described previously[25]. Binding events were defined as those that persisted for a minimum of ~2.5 s. Motile events were defined as binding events in which unidirectional movement of fluorescently labeled assay components could be observed for a minimum of five pixels on the position axis (x axis) of the kymograph. Motor run lengths and mean velocities were extracted from kymographs by measuring the total observed displacement on the position axis (x axis) or the inverse of its average slope, respectively. Analysis was done on spatially isolated microtubules on which individual runs could faithfully be tracked. Microtubules that showed signs of bundling were excluded because this can confound analysis of binding frequency and directionality. All control and experimental conditions for each independent experiment were performed on the same day; different independent experiments used fresh protein assemblies and were typically performed on separate days. Three to four fields of view in one to three imaging chambers per condition were analyzed in each experiment. Statistical analysis was performed on aggregated data from all replicates for a given condition using GraphPad Prism v.9.1.1 for MacOS (GraphPad Software). For data sets that were compared by parametric statistical tests, the normality of the data was assumed, but not explicitly tested.

## Microtubule-binding assays

For microtubule-binding assays[72], microtubules were polymerized from 5 mg ml$^{-1}$ porcine tubulin (Cytoskeleton) at 37 °C for 20 min in BRB80 supplemented with 1 mM GTP and then diluted 1:10 in BRB80 + 20 µM taxol. Proteins of interest were incubated at the indicated concentrations with 20 µl of this microtubule solution for 30 min at room temperature in a final volume of 50 µl, before soluble and microtubule-bound fractions were segregated by ultracentrifugation at 80,000g for 30 min over a 500-µl 30% sucrose cushion. Soluble fractions were supplemented with 1× SDS-loading dye. Microtubule pellets were washed with BRB80 + 20 µM taxol before redissolving in 50 µl of 1× SDS-loading dye. Soluble and pellet fractions were analyzed by SDS–PAGE.

## GST-pulldown competition assay

To test whether Tm1 competes with the Khc motor-tail interaction, 5 µM GST-Khc 855–975 as bait and an indicated excess of prey protein Khc 1–365 were incubated with increasing concentrations of Tm1 1–335 as competitor for Khc 855–975 binding in a total volume of 100 µl in GF150 buffer (25 mM HEPES/KOH pH 7.3, 150 mM KCl, 1 mM MgCl$_2$, 2 mM DTT). After incubation for 40 min on ice, the mixture was added to 50 µl of pre-equilibrated glutathione Sepharose 4B beads (GE Healthcare) and proteins were allowed to bind for 2 h at 4 °C on a nutating shaker. Beads were washed with 4× 500 µl and 1× 100 µl of GF150 + 0.01% NP-40. Bound proteins were eluted from the beads by boiling in 1× SDS-loading dye for 5 min at 95 °C. Input, wash and elution samples were analyzed by SDS–PAGE.

## Mass photometry

Mass photometry measurements were performed using a TwoMP mass photometer (Refeyn) with coexpressed, copurified Khc–Tm1 complexes at a final concentration of 20 nM in PBS. One-minute videos with a fixed image size were recorded using the AcquireMP software (v.2.5.0; Refeyn). Data analysis was performed using the DiscoverMP software (v.2.5.0; Refeyn).

## Nuclear magnetic resonance spectroscopy

NMR measurements were performed at 298 K on a Bruker Avance III NMR spectrometer with magnetic field strengths corresponding to proton Larmor frequencies of 600 MHz equipped with a cryogenic triple-resonance gradient probe head. NMR sample concentration was 50 µM for titration experiments and 177 µM for backbone assignment experiments. Experiments for backbone assignments were performed on $^{13}$C,$^{15}$N-labeled samples using conventional triple-resonance experiments (HNCO, HNCA, CBCA(CO)NH, HN(CO)CA and HNCACB)[73]. All spectra were acquired using the apodization weighted sampling scheme[74] and processed using NMRPipe v.10.9 (ref. [75]). Resonance assignments were carried out using the program Cara v.1.9.1 (http://cara.nmr.ch).

For titrations, U10 RNA (Integrated DNA Technologies) and Tm1 were added to $^{15}$N-labeled Khc 855–975 at the indicated ratios and a $^1$H-$^{15}$N HSQC was recorded for each titration point. Peak intensity ratios were derived using NMRView v.5.0.4 (ref. [76]) and corrected for dilution. The extent of amide $^1$H-$^{15}$N chemical shift perturbations in free versus bound Khc 855–975 were calculated according to Williamson[77] to compensate for frequency range differences between $^{15}$N and $^1$H dimensions. Errors of individual measurements were estimated as the standard deviation of the noise in a spectral region without any signals. Error bars of intensity ratios were calculated according to the rule of propagation of errors.

## Small-angle X-ray scattering data acquisition and structure modeling

SAXS data on Khc FL in isolation and the copurified complex of Khc FL and Tm1 FL in 25 mM HEPES/KOH pH 7.3, 150 mM KCl, 1 mM MgCl$_2$ and 2 mM DTT at concentrations of 0.5, 1.0 and 2.0 mg ml$^{-1}$ were recorded at 20 °C on the BM29 beamline (European Synchrotron Radiation Facility, Grenoble, France) (Supplementary Table 5). Ten frames with an exposure time of 1 s per frame were recorded in batch mode using an X-ray wavelength of 0.992 Å and a sample distance of 2.81 m to the Pilatus2M detector. Data reduction and buffer subtraction was performed by the automatic processing pipeline[78]. The $R_g$ and scattering intensity at zero angle ($I(0)$) were determined using the Guinier approximation using Primus/autorg[79].

The scattering curve was subsequently compared with pools of structures generated by connecting AlphaFold2 (refs. [45],[46]) models of the Khc head domain (residues 1–378) homodimer connected to four stretches of coiled-coil homodimers (residues 403–585, 604–706, 711–838 and 844–936, respectively). The linker regions between the structured parts were modified by random phi/psi rotations in monomer A of the homodimer followed by a short four-step simulated annealing energy minimization to restore proper bond geometries in monomer B, for which all corresponding residues were rotated together with monomer A during the linker randomization. The C termini of monomer A and monomer B were subsequently randomized independently. A total of 5,000 structures were generated in this way. In a second step, the structure of the motor homodimer was replaced by a homodimer (residues 1–344) bound to a peptide from the C terminus of molecule A (residues 934–953) containing the IAK motif, which was followed by simulated annealing energy minimization to restore proper bond geometry. During the simulated annealing energy minimization steps, the structures of the motor and coiled-coil models or motor/peptide were unaltered by using a harmonic energy term to keep the coordinates close to the starting coordinates in the template structure[80]. The calculations were performed using CNS v.1.2 (ref. [81]) within the ARIA v.1.2 framework[82]. Structure pools for the Khc homodimer in complex with Tm1 were generated in a similar way with an AlphaFold2-modeled α-helix of Tm1 (residues 258–386) bound to the last coiled-coil stretch of Khc (residues 844–936) in a position derived from the X-ray structure of the Khc–Tm1 complex[32] and additional randomization of all residues outside the α-helix of Tm1. Each of the 2 × 5,000 models was fitted to the SAXS scattering intensities using CRYSOL v.2.8.3 (ref. [79]).

## Circular dichroism spectroscopy

CD measurements were performed with Khc 855–975 at a concentration of 0.41 mg ml$^{-1}$ in 20 mM NaPO$_4$ pH 6.5 and 150 mM NaCl. The spectrum was recorded between 260 nm and 190 nm at 20 °C using a Jasco J-815 CD spectrometer with a 50 nm min$^{-1}$ scan speed, digital integration time of 1 s with a 1 nm bandwidth and 10 accumulations. The spectrum was analyzed with the CDSSTR[83], Contin-LL[84] and Selcon3 (ref. [85]) methods with the protein reference data set 7 (ref. [86]) using the Dichroweb server[87]. The estimation of protein secondary structure fractions from the three analysis methods were averaged.

## Reporting summary

Further information on research design is available in the Nature Portfolio Reporting Summary linked to this article.

## Data availability

The backbone chemical shift assignment of Khc 855–975 has been deposited to the BMRB under the accession code 51888 and the SAXS data have been submitted to SASBDB (IDs SASDRD8 and SASDRE8). Source data are provided with this paper. All other data is available upon request.

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

## Acknowledgements

We thank L. Dimitrova-Paternoga (European Molecular Biology Laboratory (EMBL) Heidelberg) for reagents, the EMBL Protein

Expression and Purification Core Facility, the EMBL Advanced Light Microscopy Facility, especially M. Lampe, and the EMBL Chemical Biology Core Facility, especially D. Will, for their support. We thank the Bio-SAXS beamline at European Synchrotron Radiation Facility (ESRF) Grenoble, BM29. We thank P. Pernot (ESRF), J. Kieffer (ESRF) and C. Jeffries (EMBL Hamburg) for discussions. S.H. was supported by the EMBL Interdisciplinary Postdoctoral fellowship (EIPOD) Programme under Marie Curie Cofund Actions MSCA-COFUND-FP (grant no. 664726) and Deutsche Forschungsgemeinschaft (DFG)-Forschergruppe 2333 grant (grant no. EP37/4-1) to A.E. The graphics in Fig. 7 were generated by E. Chiang as part of the MRC Laboratory of Molecular Biology's VisLab, and adapted for use in Fig. 1b. Work in the laboratories of J.H. and A.E. was supported by funding from the DFG via the priority program SPP1935 to J.H. and A.E. (grant nos EP37/3-1 and EP37/3-2) and the EMBL. Work in S.L.B.'s group is supported by the Medical Research Council (MRC), as part of United Kingdom Research and Innovation (also known as UK Research and Innovation; MRC file reference no. MC_U105178790). M.A.M. is supported by a project grant from the Biotechnology and Biological Sciences Research Council (grant no. BB/T00696X/1) awarded to S.L.B. The funders had no role in study design, data collection and analysis, decision to publish or preparation of the manuscript. For the purpose of the MRC open access policy, the authors have applied a CC-BY public copyright license to any Author Accepted Manuscript version arising.

## Author contributions

S.H., M.A.M., A.E. and S.L.B. conceived the study. S.H., M.A.M., B.S., E.M. and K.L. performed the experiments. S.H., M.A.M., B.S. and J.H. analyzed the data. S.L.B. and A.E. supervised the study. S.H. drafted the manuscript. All authors contributed to the writing of the manuscript.

## Funding

## Competing interests

The authors declare no competing interests.

## Additional information

**Extended data** is available for this paper at https://doi.org/10.1038/s41594-024-01212-x.

**Correspondence and requests for materials** should be addressed to Simon L. Bullock or Anne Ephrussi.

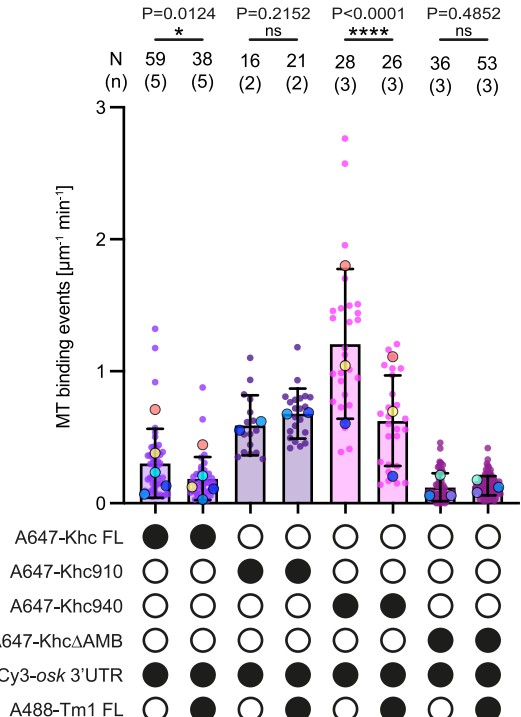

**Extended Data Fig. 1 | Effects of Khc tail truncations on motor association with microtubules.** Frequency of all microtubule (MT)-binding events (processive, static and diffusive) for Khc FL, Khc910, Khc940 and KhcΔAMB with or without Tm1 FL. For all plots, mean ± s.d. is shown and is derived from 16-59 microtubules (N) from 4–15 imaging chambers and 2–5 independent experiments (n) per condition (see Supplementary Table 1). Mean values for each independent experiment (large dots) are superimposed on values for individual microtubules (small dots). Black or white circles indicate presence or absence of indicated components, respectively. Parallel experiments are shown with large dots of the same color. Cy3-*osk* 3′ UTR was included in all experiments. Statistical significance was determined by unpaired two-tailed t-tests using N values (total number of microtubules). ns: not significant (p > 0.05).

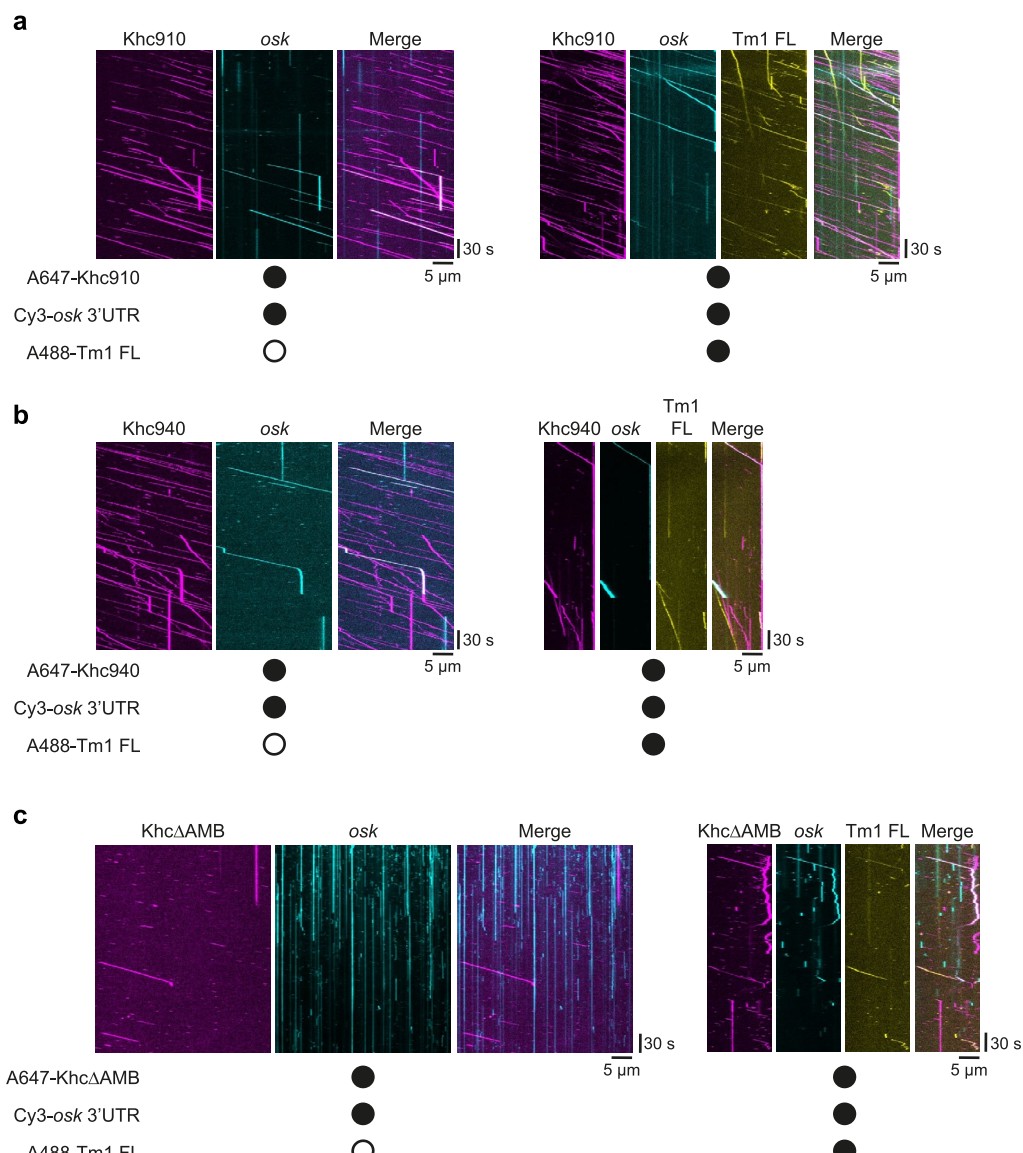

**Extended Data Fig. 2 | Tm1 inhibits Khc motility via Khc's regulatory tail.**
**a-c**, Kymographs showing motile behavior of Khc910 (**a**), Khc940 (**b**) or KhcΔAMB (**c**) and *osk* 3′ UTR in the presence or absence of Tm1 FL. Microtubule plus and minus ends are oriented toward the right and left of each kymograph, respectively. Black or white circles indicate presence or absence of indicated components, respectively.

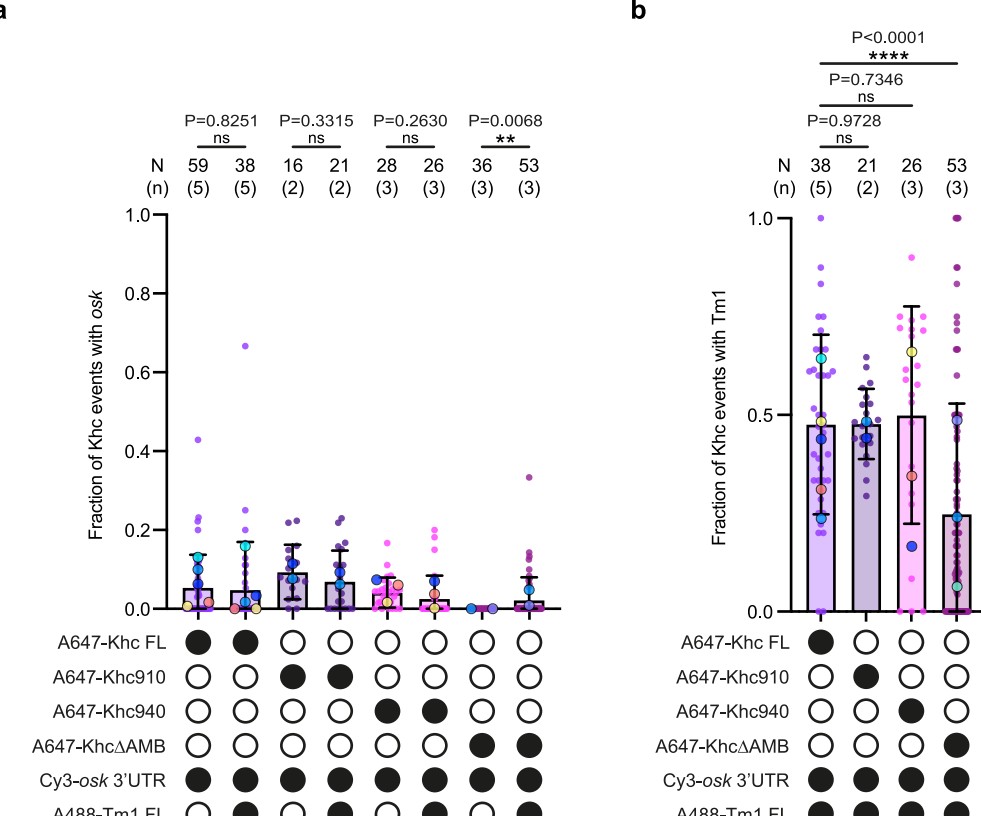

**Extended Data Fig. 3 | Effects of Khc tail truncations on motor association with *osk* RNA and Tm1. a, b,** Fraction of Khc FL, Khc910, Khc940 and KhcΔAMB microtubule-binding events that co-localized with *osk* 3′ UTR in absence or presence of Tm1 FL (**a**) or that localized with Tm1 FL (**b**). For all plots, mean ± s.d. is shown and is derived from 16–59 microtubules (N) from 4–15 imaging chambers and 2–5 independent experiments (n) per condition (see Supplementary Table 1). Mean values for each independent experiment (large dots) are superimposed on

values for individual microtubules (small dots). Black or white circles indicate the presence or absence of indicated components, respectively. Parallel experiments within the figure are shown with large dots of the same color. Cy3-*osk* 3′ UTR was included in all experiments. Statistical significance was determined by unpaired two-tailed t-tests using N values (total number of microtubules). ns: not significant (p > 0.05).

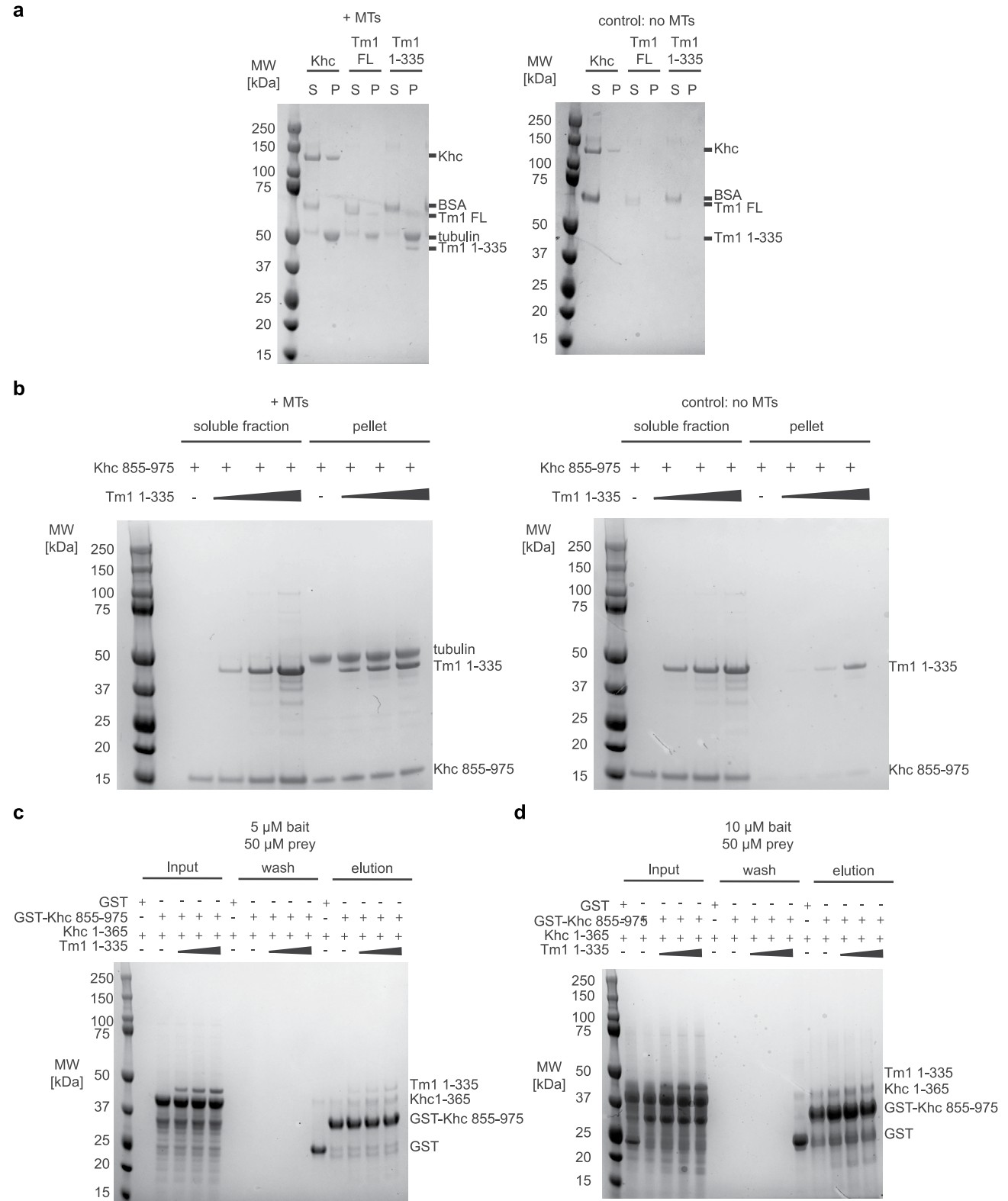

**Extended Data Fig. 4 | See next page for caption.**

**Extended Data Fig. 4 | Evaluation of the effect of Tm1 on the Khc tail's interaction with microtubules and the Khc motor domain. a**, Microtubule binding assay with Tm1 FL and Tm1 1–335. Khc FL served as a positive control. Both Tm1 FL and Tm1 1–335 partially partitioned into the microtubule pellet fraction. 10 μg of each protein was incubated with polymerized microtubules prior to segregation of microtubule-bound and -unbound fractions by ultracentrifugation over a 30% sucrose cushion. BSA was added to all samples as a negative control. To control for microtubule-independent pelleting of the proteins, controls without microtubules were performed (right). Supernatant and microtubule pellet fractions were analyzed by SDS-PAGE. **b**, Microtubule binding assay with Khc 855–975 and Tm1 1–335. Khc 855–975 partially partitioned into the microtubule pellet both in the presence and absence of increasing concentrations of Tm1 1–335. 25 μM Khc 855–975 was incubated with Tm1 1–335 at ratios of 1:0.5, 1:1 and 1:2 in the presence of polymerized microtubules before proceeding as described above. Controls for microtubule-independent pelleting of the proteins without microtubules are shown on the right. Representative images from n = 2 experiments are shown. **c, d**, GST-pulldown competition assays to assess the effect of Tm1 on Khc tail-motor interaction using 5 μM (**c**) or 10 μM (**d**) GST-Khc 855–975 as bait and 50 μM Khc 1–365 as prey. Tm1 1–335 was added at increasing concentrations (5 to 15 μM) to evaluate competition for binding to GST-Khc 855–975. Representative images from n = 2 experiments are shown.

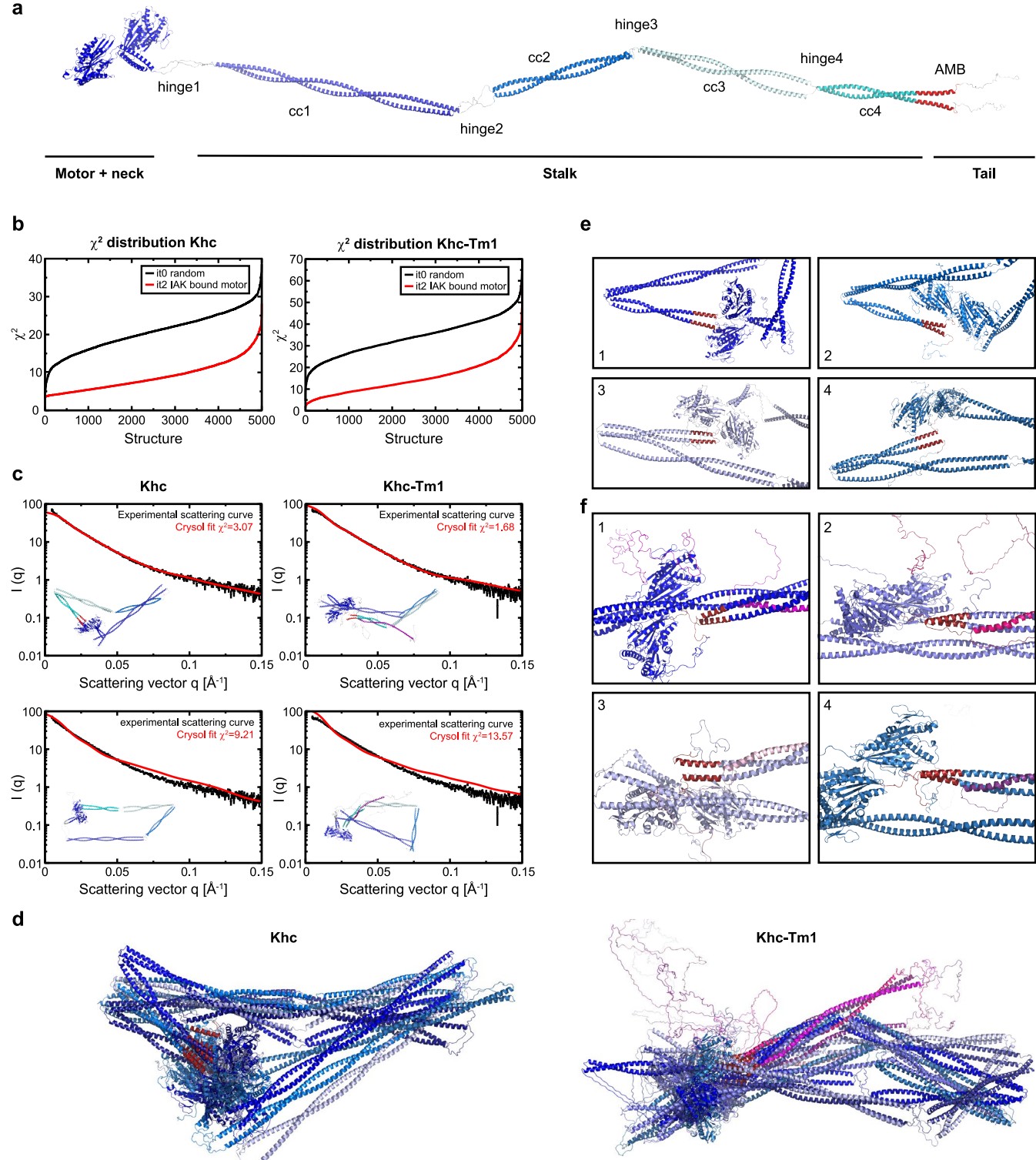

**Extended Data Fig. 5 | Additional information on SAXS-based Khc modeling.**
**a**, Template model of Khc used for randomization. Coordinates of the structured parts (motor domain: residues 1–378; four coiled-coil regions: residues 403–585, 604–706, 711–838 and 844–936) were generated by AF2[45,46] and connected with the coordinates of the missing linker residues. **b**, $\chi^2$ distribution of all Crysol fits to the Khc (left) and Khc-Tm1 (right) scattering curves. **c**, Best (upper panel) and mediocre (lower panel) fits of Khc (left) and Khc-Tm1 (right) structural models to the experimental scattering curves. **d**, Alignments of the five best fits for Khc (left) and for Khc-Tm1 (right). **e**, Close-ups of the AMB domains in the four best fitting models for Khc. **f**, Close-ups of the AMB domains in the four best fitting models for Khc-Tm1. Khc models are shown in shades of blue with the AMB domains marked in red; Tm1 is shown in shades of pink.

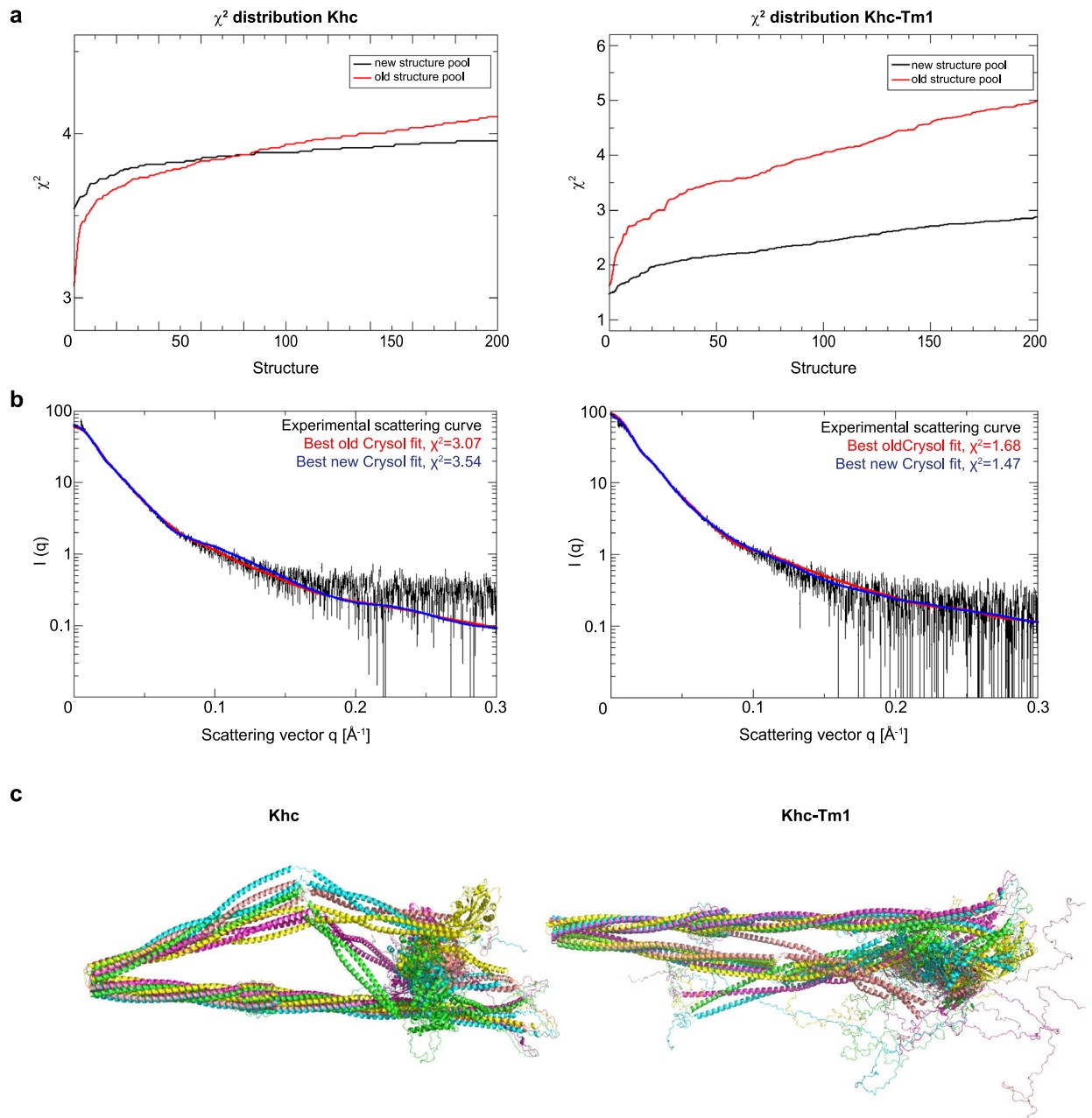

**Extended Data Fig. 6 | SAXS data fitting with models containing a rigid hinge2 region of the stalk. a**, $\chi^2$ distribution of all Crysol fits from our previous structure pools and the new structure pools with hinge2 conformation according to references[47,48] to the Khc (left) and Khc-Tm1 (right) scattering curves. **b**, Best fits of Khc (left) and Khc-Tm1 (right) structural models from our previous structure pools and the new structure pools to the experimental scattering curves. **c**, Alignments of the five best fits for Khc (left) and for Khc-Tm1 (right) from the new structure pools.

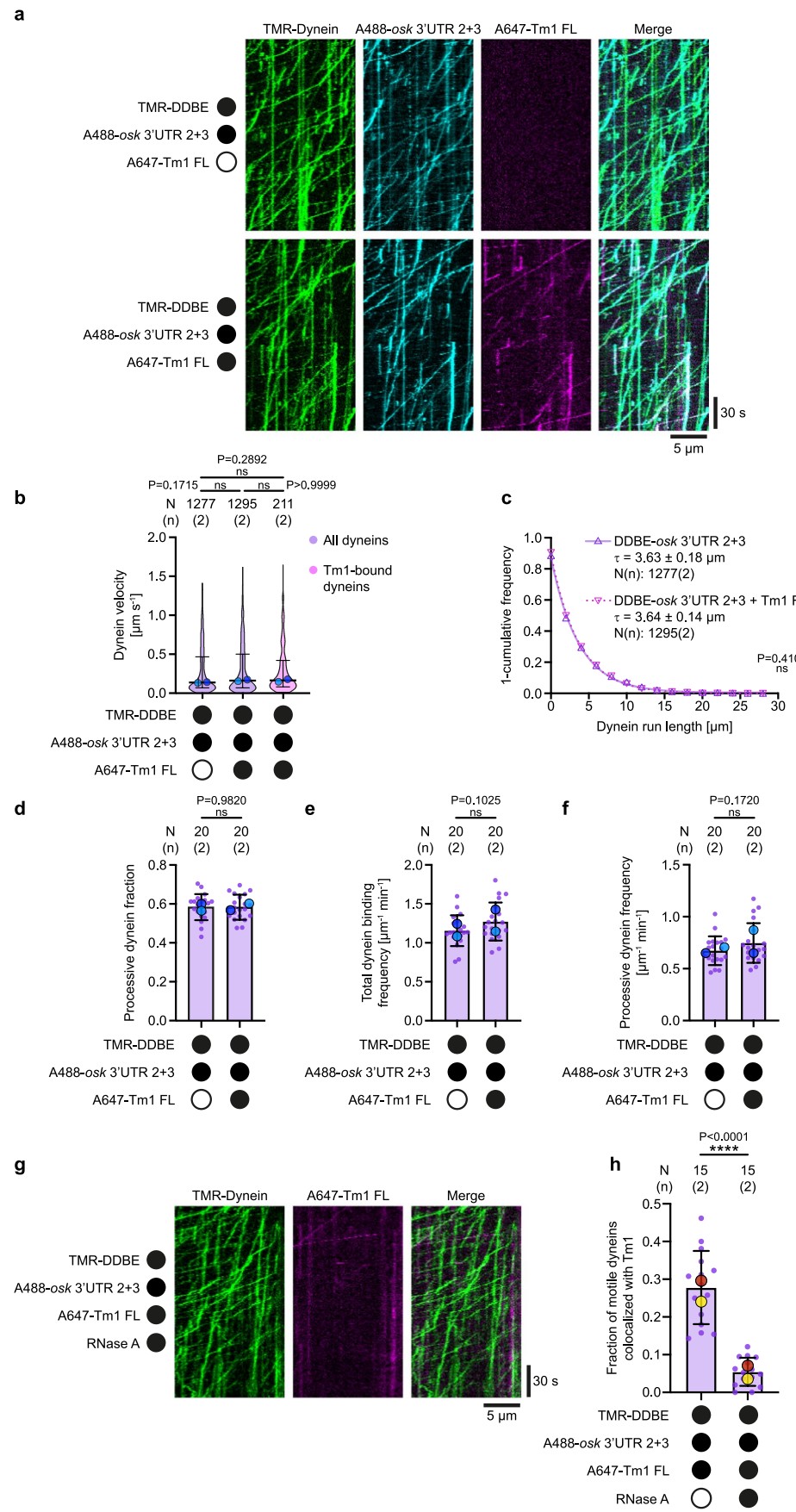

**Extended Data Fig. 7 | See next page for caption.**

**Extended Data Fig. 7 | Tm1 does not affect dynein activity in *osk* RNPs.**
**a**, Kymographs showing motile behavior of DDBE (via tetramethylrhodamine (TMR)-dynein) and *osk* 3′ UTR 2 + 3 with or without unlabeled Tm1 FL. Co-translocation of Tm1 is observed when present in the assay. **b, c**, Velocity (**b**) and run length (**c**) of DDBE-*osk* 3′ UTR 2 + 3 complexes assembled with or without Tm1. In **b**, median ± interquartile range is shown and is derived from analysis of 211–1,295 individual complexes (N) from 4 imaging chambers and 2 independent experiments (n) per condition (see Supplementary Table 3). Median values for each independent experiment (large dots) are superimposed on a violin plot showing distribution of values from individual complexes. In **c**, 1-Cumulative frequency distributions of measured run lengths were fit as one-phase decays with the decay constant τ ± 95% confidence interval shown. Hollow triangles represent empirical bin values used for fitting derived from analysis of 1,277–1,295 individual complexes (N) from 2 independent experiments (n) per condition (see Supplementary Table 3). **d-f**, Fraction of dynein binding events that underwent processive motility (**d**) and frequencies of dynein binding to microtubules (**e**) and processive dynein movement (**f**) in the presence or absence of Tm1 FL. Mean ± s.d. is shown and is derived from 20 microtubules (N), representing analysis of 2,152–2,207 individual complexes

from 3-4 imaging chambers and 2 independent experiments (n) per condition (see Supplementary Table 3). **g**, Kymographs showing reduced association of Tm1 with motile DDBE-*osk* 3′ UTR 2 + 3 RNPs after RNAse A treatment. **h**, Fraction of moving DDBE-*osk* 3′ UTR 2 + 3 RNPs that co-localized with Tm1 signal with or without RNase A. Mean ± s.d. is shown and is derived from 15 microtubules (N), representing analysis of 559–886 individual complexes from 2-3 imaging chambers and 2 independent experiments (n) per condition. Black or white circles indicate presence or absence of indicated components, respectively. In **a** and **g**, microtubule plus and minus ends are oriented toward the right and left of each kymograph, respectively. In panels **b, d-f** and **h**, mean values for each independent experiment (large dots) are superimposed on violin plots showing the distribution of values from individual complexes (**b**) or on values for individual microtubules (small dots; **d-f, h**); parallel experiments within each of these panels are shown by large dots of the same color. Statistical significance was determined by a nonparametric one-way Kruskal-Wallis ANOVA test with Dunn's test for multiple comparisons (**b**), unpaired two-tailed Mann-Whitney test (**c**), unpaired two-tailed t-tests (**d-f**), or unpaired two-tailed t-test with Welch's correction (**h**) using N values (total number individual complexes (**b, c**) or total number of microtubules (**d-f, h**)). ns: not significant (p > 0.05).

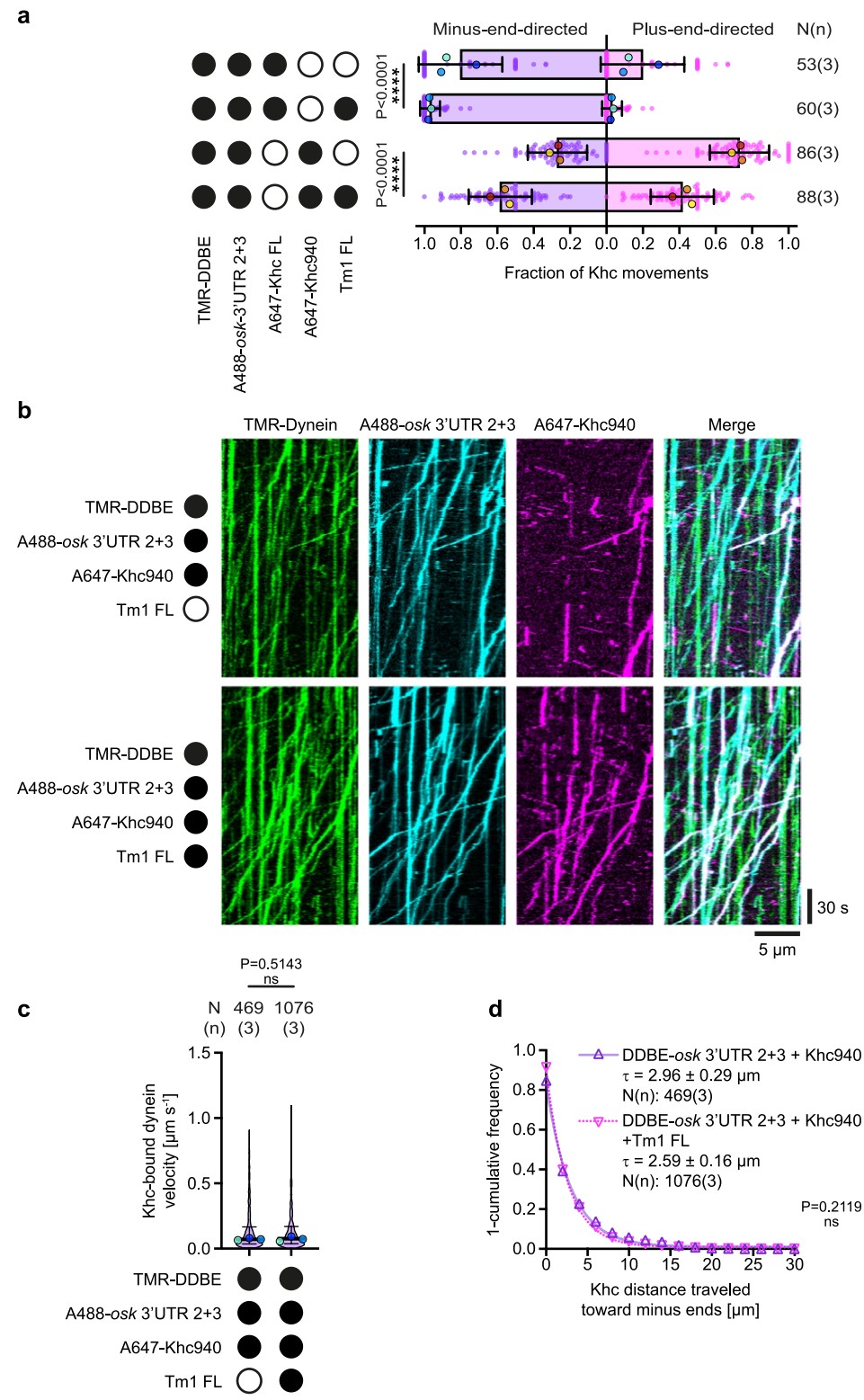

**Extended Data Fig. 8 | See next page for caption.**

**Extended Data Fig. 8 | Motile properties of DDBE-*osk* with Khc940 in the presence and absence of Tm1. a**, Fraction of processive Khc FL and Khc940 movements that moved towards microtubule minus ends (towards left of x-axis, purple) and plus ends (towards right of x-axis, magenta) with or without Tm1 FL. The mean ± s.d. is shown and is derived from 53–88 microtubules (N), representing analysis of 178–1,795 individual complexes from 5-6 imaging chambers and 3 independent experiments (n) per condition (See Supplementary Table 4). Mean values for each independent experiment (large dots) are superimposed on values for individual microtubules (small dots). **b**, Kymographs showing motile behavior of DDBE (via TMR-dynein), Khc940, and *osk* 3′ UTR 2 + 3 with or without Tm1 FL. Microtubule plus and minus ends are oriented toward the right and left of each kymograph, respectively. **c**, Velocity of minus-end-directed DDBE-*osk* 3′ UTR 2 + 3 complexes with co-migrating Khc940 signal with or without Tm1 FL. The median ± interquartile range is shown and is derived from analysis of 469–1,076 individual complexes (N) from 5-6 imaging chambers

and 3 independent experiments (n) per condition (see Supplementary Table 4). Median values for each independent experiment (large dots) are superimposed on a violin plot showing distribution of values from individual complexes. **d**, Distance traveled by Khc940 towards the minus end in the presence and absence of Tm1 FL. 1-Cumulative frequency distributions of measured run lengths were fit as one-phase decays with the decay constant τ ± 95% confidence interval shown. Hollow triangles represent empirical bin values used for fitting derived from analysis of 469–1,076 individual complexes (N) from 3 independent experiments (n) per condition (see Supplementary Table 4). Black or white circles indicate presence or absence of indicated components, respectively. In panels **a** and **c**, parallel experiments are shown with large dots of the same color. Statistical significance was determined by an unpaired two-tailed t-test and and t-test with Welch's correction (**a**) or unpaired two-tailed Mann-Whitney test (**c**, **d**) using N values (total number of microtubules (**a**) or total number of individual complexes (**c**, **d**). ns: not significant (p > 0.05).

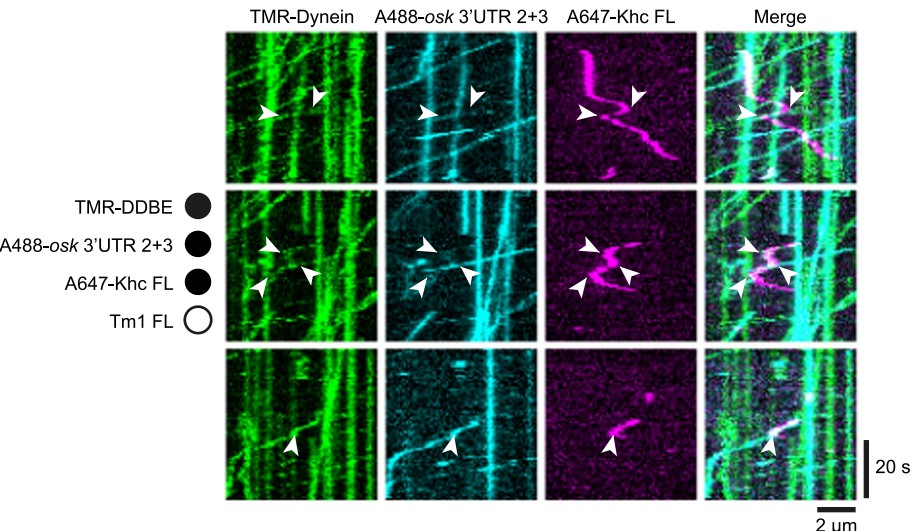

**Extended Data Fig. 9 | Khc exhibits rare directional switches related to the binding and dissociation of DDBE in the absence of Tm1.** Kymographs showing instances of bidirectional Khc FL movements in the absence of Tm1 FL. Directional switches of Khc (white arrowheads) correlate with DDBE-*osk* 3' UTR 2 + 3 association and dissociation events. Microtubule plus and minus ends are oriented toward the right and left of each kymograph, respectively. Black or white circles indicate presence or absence of indicated components, respectively.

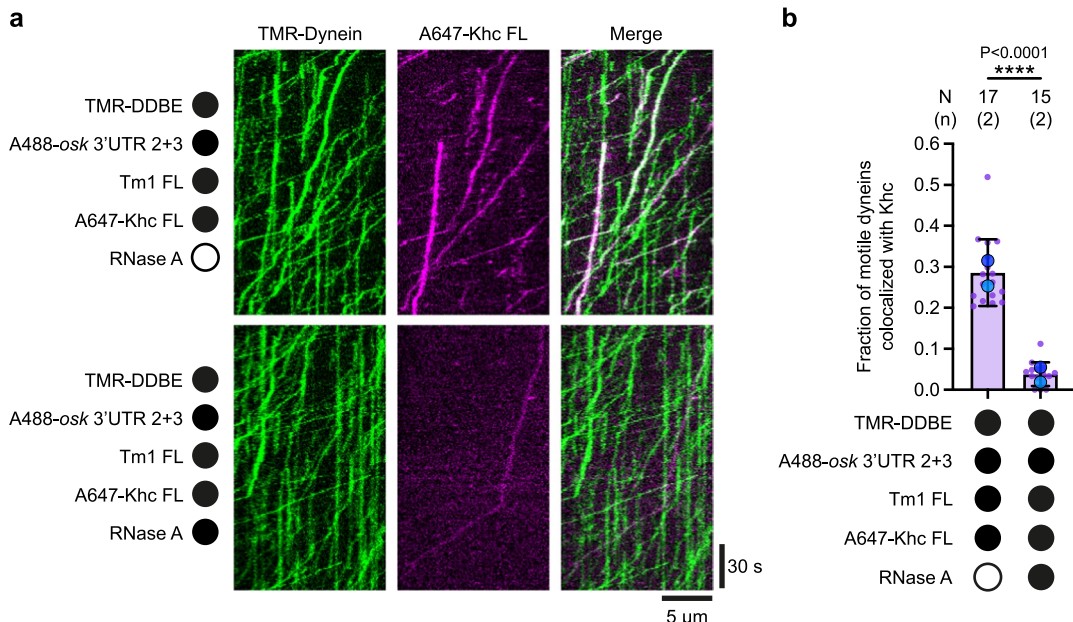

**Extended Data Fig. 10 | Tm1-induced association of Khc with DDBE is dependent on RNA. a**, Kymographs showing reduced association of Khc FL with motile DDBE-*osk* 3′ UTR 2 + 3 RNPs in the presence of Tm1 FL after treatment with RNase A. Microtubule plus and minus ends are oriented toward the right and left of each kymograph, respectively. **b**, Fraction of moving DDBE-*osk* 3′ UTR 2 + 3 RNPs that colocalized with Khc FL signal with and without RNase A treatment in the presence of Tm1 FL. The mean ± s.d. is shown and is derived from 15–17 microtubules (N), representing analysis of 963–1,079 individual complexes from 4 imaging chambers and 2 independent experiments (n) per condition. Mean values for each independent experiment (large dots) are superimposed on values for individual microtubules (small dots). Black or white circles indicate the presence or absence of indicated components, respectively. Parallel experiments are shown using large dots of the same color. Statistical significance was determined by an unpaired two-tailed t-test with Welch's correction using N values (total number of microtubules).

# Reporting Summary

## Statistics

For all statistical analyses, confirm that the following items are present in the figure legend, table legend, main text, or Methods section.

| n/a | Confirmed | |
|---|---|---|
| ☐ | ☒ | The exact sample size (*n*) for each experimental group/condition, given as a discrete number and unit of measurement |
| ☐ | ☒ | A statement on whether measurements were taken from distinct samples or whether the same sample was measured repeatedly |
| ☐ | ☒ | The statistical test(s) used AND whether they are one- or two-sided *Only common tests should be described solely by name; describe more complex techniques in the Methods section.* |
| ☐ | ☒ | A description of all covariates tested |
| ☐ | ☒ | A description of any assumptions or corrections, such as tests of normality and adjustment for multiple comparisons |
| ☐ | ☒ | A full description of the statistical parameters including central tendency (e.g. means) or other basic estimates (e.g. regression coefficient) AND variation (e.g. standard deviation) or associated estimates of uncertainty (e.g. confidence intervals) |
| ☐ | ☒ | For null hypothesis testing, the test statistic (e.g. *F*, *t*, *r*) with confidence intervals, effect sizes, degrees of freedom and *P* value noted *Give P values as exact values whenever suitable.* |
| ☒ | ☐ | For Bayesian analysis, information on the choice of priors and Markov chain Monte Carlo settings |
| ☒ | ☐ | For hierarchical and complex designs, identification of the appropriate level for tests and full reporting of outcomes |
| ☒ | ☐ | Estimates of effect sizes (e.g. Cohen's *d*, Pearson's *r*), indicating how they were calculated |

*Our web collection on statistics for biologists contains articles on many of the points above.*

## Software and code

Policy information about availability of computer code

| Data collection | Leica LAS X, Micromanager<br>AcquireMP (Refeyn Ltd; version 2.5.0)<br>Topspin 3.5 (Bruker)<br>ESRF BM29 automatic processing pipeline |
|---|---|
| Data analysis | FIJI (ImageJ2 version 2.9.0/1.53t)<br>GraphPad Prism version 9.1.1<br>DiscoverMP (Refeyn Ltd; version 2.5.0)<br>NMRPipe v10.9, Cara 1.9.1, NMRView 5.0.4<br>ATSAS 3.0.4, CNS-1.2<br>CRYSOL 2.8.3 |

For manuscripts utilizing custom algorithms or software that are central to the research but not yet described in published literature, software must be made available to editors and reviewers. We strongly encourage code deposition in a community repository (e.g. GitHub). See the Nature Portfolio guidelines for submitting code & software for further information.

## Data

Policy information about availability of data

All manuscripts must include a data availability statement. This statement should provide the following information, where applicable:
- Accession codes, unique identifiers, or web links for publicly available datasets
- A description of any restrictions on data availability
- For clinical datasets or third party data, please ensure that the statement adheres to our policy

> Source data are provided with this paper. The backbone chemical shift assignment of Khc 855-975 has been deposited to the BMRB under the accession code 51888 and the SAXS data have been submitted to SASBDB (IDs SASDRD8 and SASDRE8).

## Human research participants

Policy information about studies involving human research participants and Sex and Gender in Research.

| | |
|---|---|
| Reporting on sex and gender | n/a |
| Population characteristics | n/a |
| Recruitment | n/a |
| Ethics oversight | n/a |

Note that full information on the approval of the study protocol must also be provided in the manuscript.

# Field-specific reporting

Please select the one below that is the best fit for your research. If you are not sure, read the appropriate sections before making your selection.

☒ Life sciences　　　☐ Behavioural & social sciences　　　☐ Ecological, evolutionary & environmental sciences

For a reference copy of the document with all sections, see nature.com/documents/nr-reporting-summary-flat.pdf

# Life sciences study design

All studies must disclose on these points even when the disclosure is negative.

| | |
|---|---|
| Sample size | Typical experiments yielded dozens to hundreds of single-molecule observations from dozens of individual microtubules. These experiments were independently replicated at least twice and with a greater number of replicates in the vast majority of replicates. |
| Data exclusions | Poorly resolved microtubules or microtubule bundles were not analyzed as they obscure and distort motor metrics. Apparent interactions of labeled protein/RNA with microtubules were excluded from analysis if they failed to meet the defining criteria for any given metric as defined in Methods. |
| Replication | All results were replicated both within the paper using identical experimental conditions and independently under varying conditions not described in the study. |
| Randomization | Overall, randomization was not applicable as biophysical metrics were gathered for individual molecules/purified components. To prevent weighting of these metrics in favor of replicates that stochastially had more microtubules in a field of view than others, microtubules in more densely coated cover slips were randomly selected for analysis prior to viewing the behavior of motor components. |
| Blinding | NMR and SAXS data of purified proteins cannot be blinded as they are measuring innate properties of these components and not different between two experimental conditions. For single-molecule experiments, fields view and microtubules for imaging were selected before viewing channels for other labeled protein/RNA components. |

# Reporting for specific materials, systems and methods

We require information from authors about some types of materials, experimental systems and methods used in many studies. Here, indicate whether each material, system or method listed is relevant to your study. If you are not sure if a list item applies to your research, read the appropriate section before selecting a response.

## Materials & experimental systems

| n/a | Involved in the study |
|---|---|
| ☒ | ☐ Antibodies |
| ☐ | ☒ Eukaryotic cell lines |
| ☒ | ☐ Palaeontology and archaeology |
| ☒ | ☐ Animals and other organisms |
| ☒ | ☐ Clinical data |
| ☒ | ☐ Dual use research of concern |

## Methods

| n/a | Involved in the study |
|---|---|
| ☒ | ☐ ChIP-seq |
| ☒ | ☐ Flow cytometry |
| ☒ | ☐ MRI-based neuroimaging |

## Eukaryotic cell lines

Policy information about cell lines and Sex and Gender in Research

| | |
|---|---|
| Cell line source(s) | Sf21 (ECACC 05022801): Derived from pupal ovarian tissue of Spodoptera frugiperda<br>Sf9 (ECACC 89070101): Derived from pupal ovarian tissue of Spodoptera frugiperda |
| Authentication | The cell lines were not authenticated. |
| Mycoplasma contamination | Insect cell lines used for production of recombinant proteins were not tested for Mycoplasma contamination. |
| Commonly misidentified lines<br>(See ICLAC register) | No commonly misidentified cell lines were used in the study. |

