## [Peer Review File · Nature Structural & Molecular Biology]

Peer Review Information

Manuscript Title: Tropomyosin 1-I/C co-ordinates kinesin-1 and dynein motors during oskar mRNA transport

Corresponding author name(s): Anne Ephrussi, Simon Bullock

Redactions – unpublished data Parts of this Peer Review File have been redacted as indicated to maintain the confidentiality of unpublished data.

Reviewer Comments & Decisions:

Decision Letter, initial version:

Message: 15th May 2023

Dear Dr. Ephrussi,

Thank you again for submitting your manuscript "Tropomyosin 1-I/C co-ordinates kinesin-1 and dynein motors during oskar mRNA transport". I apologise for the delay in responding, which resulted from the difficulty in obtaining suitable referee reports. Nevertheless, we now have comments (below) from the 3 reviewers who evaluated your paper. In light of those reports, we remain interested in your study and would like to see your response to the comments of the referees, in the form of a revised manuscript.

You will see that all referees appreciate the findings as interesting and their potential for progressing the field's conceptual understanding of how particular cargos are transported by specific motors. There are, however, a few very important issues that should be addressed in a revision. First and foremost, reviewer #3 raises concerns about the robustness of the findings, given the limited number of measurements and repeats, which need to be fully addressed by performing the requested repeats and expanding analyses. Moreover, we ask you to please follow the guidelines of reviewer #1 with respect to additional experiments that would solidify the reached mechanistic conclusions (e.g. repeats in the presence of ensconsin, repeat assays using the Khc940 construct, etc) and of reviewer #3 with respect to validating certain findings by performing CD experiments. Finally, both reviewer #2 and reviewer #3 provide useful suggestions on how to textually improve the manuscript and further contextualise the findings with respect to existing literature that we editorially agree that would elevate its value.

Please be sure to address/respond to all concerns of the referees in full in a point-by-point

response and highlight all changes in the revised manuscript text file. If you have comments that are intended for editors only, please include those in a separate cover letter.

We expect to see your revised manuscript within 3 months. If you cannot send it within this time, please contact us to discuss an extension; we would still consider your revision, provided that no similar work has been accepted for publication at NSMB or published elsewhere.

Reporting Summary:

When submitting the revised version of your manuscript, please pay close attention to our [href="https://www.nature.com/nature-portfolio/editorial-policies/image-integrity">Digital Image Integrity Guidelines.](https://www.nature.com/nature-portfolio/editorial-policies/image-integrity) and to the following points below:

Data availability: this journal strongly supports public availability of data. All data used in accepted papers should be available via a public data repository, or alternatively, as Supplementary Information. If data can only be shared on request, please explain why in your Data Availability Statement, and also in the correspondence with your editor. Please note that for some data types, deposition in a public repository is mandatory - more information on our data deposition policies and available repositories can be found below: <https://www.nature.com/nature-research/editorial-policies/reporting-standards#availability-of-data>

[Redacted]

Sincerely,

Dimitris Typas
Associate Editor
Nature Structural & Molecular Biology

ORCID: 0000-0002-8737-1319

Referee expertise:

Referee #1: dynein biophysics

Referee #2: structural and biophysical analysis of mRNA transport

Referee #3: kinesin motors and in vitro reconstitution of motor regulation

Reviewers' Comments:

Reviewer #1:

Remarks to the Author:

This manuscript studies the RNA transport machinery in *Drosophila* cells using in vitro reconstitution and single-molecule fluorescence imaging. Previous studies have shown that oskar mRNA is transported to the oocyte by dynein in complex with dynactin, BicD, and Egl. Kinesin is linked to oskar through Tm1. Kinesin remains inactive when dynein transports oskar, is activated at later stages of oogenesis, and transports it to the posterior pole. How the two motors are linked simultaneously to oskar RNPs, and how cargo-bound kinesin is inhibited during dynein-mediated transport into the oocyte has been unclear. Using TIRF microscopy-based in vitro motility assays using fluorescently labeled versions of recombinant, full-length and truncated versions of kinesin and Tm1; dynein-dynactin-BicD-Egl (DDBE), and in vitro transcribed oskar 3'UTR RNA, the authors have shown that Tm1 inhibits kinesin motility by stabilizing its autoinhibited conformation through its interaction with the kinesin regulatory tail domain. Using NMR and SAXS, the authors obtained evidence for a Tm1-induced conformational change in the folded conformation of full-length kinesin. Consistent with the studies in live cells, they also showed that Tm1 links kinesin to the DDBE complex transporting oskar towards the minus end. They propose that Tm1 is a noncanonical light chain of kinesin that links this motor to the dynein-transported RNA and prevents tug-of-war between these opposing motors by stabilizing the autoinhibited conformation of dynein.

There is a growing interest in how motors are recruited to specific cargos and how adapter proteins that recruit motors also contribute to the regulation of bidirectional transport. In vitro reconstitution approaches are particularly powerful as they serve as an ideal platform to test specific models in the absence of other proteins. Therefore, the work presented in this manuscript is timely and broadly interesting to the intracellular transport and motor proteins community. Overall, the work is also well done. I would be supportive of the publication of this work in NSMB if the authors can adequately address my concerns listed below.

Major Concerns

1. Recent studies have shown that kinesin-1 remains autoinhibited and its run frequency is increased over 40-fold by the addition of 10-50 nM ensconsin (MAP7). Consistently, example kymographs in Figure 1 show that the run frequency of full-length kinesin is much lower than tail truncated kinesins or the DDBE complex. The authors wrote that tail-truncated kinesin constructs are hyperactive. I disagree with this view. Instead, full-length kinesin is autoinhibited and tail truncated constructs are either fully or partially rescued from autoinhibition, as suggested by the literature. This is also evident from the very low velocity of full-length kinesin compared to tail-truncated kinesin constructs. Figure 2

clearly shows that Tm1 strongly inhibits the constitutively active Khc940 construct, so there is strong evidence that Tm1 stabilizes the autoinhibited conformation. However, as is, figure 1 shows that Tm1 further inhibits a motor that is already autoinhibited. To conclusively show that Tm1 inhibits full-length kinesin-1 motility, the authors should repeat their experiment in the presence of ensconsin, which is needed to activate kinesin-1 motility. According to their conclusions, they should observe full-length kinesin to exhibit robust motility and Tm1 to reduce kinesin run frequency on ensconsin-decorated microtubules.

2. Similarly, Figure 3 shows that KhcdeltaAMB is also an autoinhibited motor. Unlike full-length kinesin, Tm1 has no effect on its motility. These experiments also need to be repeated in the presence of ensconsin to test if KhcdeltaAMB exhibits robust motility on ensconsin-decorated microtubules, and Tm1 does not inhibit its motility.

3. Figures 7 and 8 show that Tm1 links kinesin-1 to oskar transported by the DDBE complex. These assemblies almost exclusively move towards the minus end, consistent with the autoinhibition or Tm1-induced inhibition of kinesin-1. The authors also claim that Tm1-mediated inhibition prevents tug-of-war between kinesin and dynein, but I do not see strong evidence supporting this view. Example kymographs in Figure 8a show little to no colocalization of kinesin with DDBE-oskar in the absence of Tm1. There is only infrequent and transient colocalization of kinesin with DDBE. I also do not see colocalization of plus end-directed kinesin (white arrowhead) with dynein. Most likely, this kinesin walks to the plus end on its own, independent of dynein. In the presence of Tm1, kinesin exhibits robust colocalization to DDBE-oskar and is transported by dynein to the minus end. These results show that Tm1 is the adapter that links kinesin-1 to the rest of the transport complex. However, they did not convincingly demonstrate that Tm1-dependent inhibition of kinesin is important for the rapid transport of the entire complex to the minus end. Given that full-length kinesin is autoinhibited on its own, it is unlikely to reduce the velocity of the complex even in the absence of Tm1-mediated inhibition. The authors should either tune down their conclusions based on these results or present more compelling experimental evidence that Tm1-mediated inhibition is important for the minus-end directed transport of kinesin. One possible experiment would be to repeat these assays using the Khc940 construct (if it still binds to Tm1 and/or RNA), which exhibits robust motility and is strongly inhibited by Tm1.

Minor concerns

1. Figure 7 is a control experiment and does not deserve to be a main text figure on its own. I recommend combining a part of this figure with Figure 8 and relegating its detailed analysis to the supplement.

2. The analysis of the data shown in Figure 8 can be slightly misleading as is (see major comment #3 above). Kinesin does not seem to localize robustly to oskar without Tm1. Therefore, the frequency of kinesin runs colocalizing with oskar in the presence and absence of Tm1 (as this seems to be the most profound effect) needs to be shown clearly before the more detailed analysis shown in this figure.

3. Lines 359-360: The authors wrote "This work involves the first reconstitution of a native cargo-adaptor-motor complex that recapitulates regulation of both dynein and kinesin motors." I think this is an overstatement and needs to be softened. Recent studies on *in vitro* reconstitution of bidirectional transport machinery showed different types of motor regulation. For example, Fenton et al and Canty et al showed dynein transports kinesin to the minus end when both motors are recruited by the TRAK adaptor. I recommend authors briefly discuss these and other relevant studies that used a similar approach and more clearly state the differences between their system with the recent work in the field.

Reviewer #2:

Remarks to the Author:

In this manuscript Heber and co-workers discuss how the relation between kinesin and dynein, two protein motors that transport RNA cargoes (among others) with opposite polarity in the microtubules system, is established. They focus in particular on the protein Tm1, a regulator of kinesin activity, and use in vitro functional and binding assays and an ensemble of methods reporting on interactions, structure and dynamics to show that Tm-1 binding is connected to a conformational change in the interacting region of kinesin, and that this strengthen kinesin inhibition. At the same time, the regulator links the inhibited kinesin to the dynein protein-RNA complex. The work provides an initial overview of the coupling of the two motors, both at the functional and molecular level.

This is an interesting paper that provides important novel molecular insight into a complex coupling mechanism essential to mRNA local translation. The study is clearly and logically structured and the data are well presented. Conclusion are, in general, not overstated and the final model presented integrates effectively the insight gained in this study. Overall, this is a high quality paper.

There are a few issues that need to be addressed prior to publication – in particularly in the discussion of the structural data. This are important, as the mechanistic detail of the coupling is a key part of the paper.

Line 212. The discussion on short transversal relaxation times of the amino-terminal part of the Khc construct leading to the loss of the NMR signal is based, as I understand it, on the assumption that this region folds as a coil-coil. This is very likely, but should be validated, for example using CD.

Later on the page. The increase in linewidth in the N-terminal part of the resonance-assigned region of the kinesin upon Tm1 binding, with and without RNA (Figures 5c and d) seems monotonous. This would be consistent with a conformational transition spreading from the N-terminus of the protein. It would be worth the authors comment on this. Again, this increase should be checked directly, using either CD or the carbon chemical shifts of the backbone resonances of the protein. In general, these chemical shifts should be discussed.

Also on this, mentioning the IAK motif in this context, may result confusing, as the TM1 binding and inhibition do not depend on the IAK.

Line 236. The binding of the motor domain results in broadening and shifts that are not monotonous. This is consistent with the hypothesis discussed by the authors, i.e. that the binding sites of the motor region and of TM1 to the carboxyterminal part of the kinesin are proximal but largely non-overlapped.

Notably, in Figure 5G, some of the shifts caused by the binding of the motor head (say residues 953-967) are the same whether TM1 is present or not. As I understand it, this is consistent with what discussed by the authors on the location of the binding sites of the two regulatory interactions and, it seems to me, confirms TM1 does not displace the motor head. On other residues instead, changes are observed which are more difficult to interpret, likely resulting by the two combined interactions.

In general, the NMR data should be discussed more in detail. At the moment this detail is missing and the basis for the conclusions drawn are often implicit.

Line 257 and figures 6. We need more detailed information of the number and type of models. It is not very easy to judge whether this is one of several models that are compatible with the data or the only one. This should be discussed explicitly and more in detail in the paper. Even better would be some validation of the model.

When discussing the microtubules experiments, I find that the authors could explain more in detail how they interpret the images we are looking at. For example, in page 4 line 93 they mention Figure 1C shows transport of oskar RNA. One can imagine how transport is derived, but it would be useful to explain it. Just below this sentence, the authors mention the value does not change significantly and give values of 4.8% and 2.8 % for the experiment in the absence and in the presence of Tm1. It would be probably better to have also the errors here, or avoid specific numbers.

Reviewer #3:

Remarks to the Author:

This is an interesting paper showing that tropomyosin-1 links kinesin-1 in an autoinhibited state to a dynein-containing osk mRNA transport complex. The idea is that kinesin is loaded already in the nurse cells and only gets active when arriving in the oocyte where kinesin-1 takes over the transport. This is a novel and interesting concept, which would be of significant interest to the cytoskeleton and motor field.

However, I have some issues with the robustness of the science. The numbers of observations are generally very low and seem to have been derived in many cases from single experiments, often less than 10 microtubules that could possibly have been found in a single field of view. Therefore, the statistical significance is meaningless as a pipetting error alone would be enough to result in a change in binding events. Some of the kymographs provided also seem not to support the quantitative data. The velocity of KHC-containing events shown in Figure 8a seems lower than that of dynein without KHC. However, the quantification in 8d and 7b contradict this. Of course, if presence of KHC would lower the speed of dynein, this would suggest that loaded KHC is able to bind to microtubules and not being autoinhibited. Thus it is fairly important for the conclusions of the manuscript to solidify those findings.

Two recent manuscripts updated our knowledge on kinesin-1 autoinhibition: 10.1126/sciadv.abp9660 and 10.1101/2023.01.26.525761. The former is briefly mentioned, the latter not. It would be important that models compared to SAXS data here would include models based on those recent findings. The data provided in the other papers show a compact shape of KHC in the absence of TM-1. Therefore a detailed discussion why Tm-1 is required to compact kinesin-1 in the current study is absolutely required.

Therefore, I can only recommend publication after experiments have been conducted with suitable independent repeats and data are provided so that variation within each experiment and across repeats are presented. I would like to refer the authors to this for guidance: 10.1083/jcb.202001064

Author Rebuttal to Initial comments

Response to the Reviewers' comments

We are very grateful to the reviewers for evaluating our work and for their insightful and constructive feedback, which we believe has led us to improve the manuscript significantly. We include below a point-by-point response to their comments. To facilitate evaluation of our revisions, we have uploaded a version of the manuscript with significant changes highlighted in blue.

Reviewer #1:

Remarks to the Author:

This manuscript studies the RNA transport machinery in *Drosophila* cells using in vitro reconstitution and single-molecule fluorescence imaging. Previous studies have shown that oskar mRNA is transported to the oocyte by dynein in complex with dynactin, BicD, and Egl. Kinesin is linked to oskar through Tm1. Kinesin remains inactive when dynein transports oskar, is activated at later stages of oogenesis, and transports it to the posterior pole. How the two motors are linked simultaneously to oskar RNPs, and how cargo-bound kinesin is inhibited during dynein-mediated transport into the oocyte has been unclear. Using TIRF microscopy-based in vitro motility assays using fluorescently labeled versions of recombinant, full-length and truncated versions of kinesin and Tm1; dynein-dynactin-BicD-Egl (DDBE), and in vitro transcribed oskar 3'UTR RNA, the authors have shown that Tm1 inhibits kinesin motility by stabilizing its autoinhibited conformation through its interaction with the kinesin regulatory tail domain. Using NMR and SAXS, the authors obtained evidence for a Tm1-induced conformational change in the folded conformation of full-length kinesin. Consistent with the studies in live cells, they also showed that Tm1 links kinesin to the DDBE complex transporting oskar towards the minus end. They propose that Tm1 is a noncanonical light chain of kinesin that links this motor to the dynein-transported RNA and prevents tug-of-war between these opposing motors by stabilizing the autoinhibited conformation of dynein.

There is a growing interest in how motors are recruited to specific cargos and how adapter proteins that recruit motors also contribute to the regulation of bidirectional transport. In vitro reconstitution approaches are particularly powerful as they serve as an ideal platform to test specific models in the absence of other proteins. Therefore, the work presented in this manuscript is timely and broadly interesting to the intracellular transport and motor proteins community. Overall, the work is also well done. I would be supportive of the publication of this work in NSMB if the authors can adequately address my concerns listed below.

Major Concerns

1. Recent studies have shown that kinesin-1 remains autoinhibited and its run frequency is increased over 40-fold by the addition of 10-50 nM ensconsin (MAP7). Consistently, example kymographs in Figure 1 show that the run frequency of full-length kinesin is much lower than tail truncated kinesins or the DDBE complex. The authors wrote that tail-truncated kinesin constructs are hyperactive. I disagree with this view. Instead, full-length kinesin is autoinhibited and tail truncated constructs are either fully or partially rescued from autoinhibition, as suggested by the literature. This is also evident from the very low velocity of full-length kinesin compared to tail-truncated kinesin constructs.

Our initial description of Khc tail truncations as hyperactive was intended to emphasize the increases in activity we observed relative to full length Khc (Khc FL). However, we appreciate that more precise language is needed given that Khc FL can adopt an autoinhibited state. We have therefore removed statements of hyperactivity and updated the text to clarify our interpretation that truncated Khcs are more active than Khc FL because they are relieved of the autoinhibition imposed by the regulatory tail domain.

Figure 2 clearly shows that Tm1 strongly inhibits the constitutively active Khc940 construct, so there is strong evidence that Tm1 stabilizes the autoinhibited conformation. However, as is, figure 1 shows that Tm1 further inhibits a motor that is already autoinhibited. To conclusively show that Tm1 inhibits full-length kinesin-1 motility, the authors should repeat their experiment in the presence of ensconsin, which is needed to activate kinesin-1 motility. According to their conclusions, they should observe full-length kinesin to exhibit robust motility and Tm1 to reduce kinesin run frequency on ensconsin-decorated microtubules.

We have strengthened the evidence that Tm1 inhibits full length Khc motility through additional independent experiments (presented in Figure 1 and 2). Regarding Ensconsin (Ens), we previously speculated in the Discussion that this could be the factor that relieves Tm1-based inhibition of Khc in the oocyte to promote oskar mRNA delivery to the posterior (line 562-566 of the revised manuscript). Consistent with this notion, Ens protein is enriched in the oocyte relative to the nurse cells (Sung et al. 2008) (a point now added to this section of the Discussion). If our hypothesis about the role of Ens is correct, we would expect that Ens can overcome the suppression of Khc activity by Tm1 in vitro, in contrast to the alternative outcome raised by the reviewer.

2. Similarly, Figure 3 shows that KhcdeltaAMB is also an autoinhibited motor. Unlike full-length kinesin, Tm1 has no effect on its motility. These experiments also need to be repeated in the presence of ensconsin to test if KhcdeltaAMB exhibits robust motility on ensconsin-decorated microtubules, and Tm1 does not inhibit its motility.

3. Figures 7 and 8 show that Tm1 links kinesin-1 to oskar transported by the DDBE complex. These assemblies almost exclusively move towards the minus end, consistent with the autoinhibition or Tm1-induced inhibition of kinesin-1. The authors also claim that Tm1-mediated inhibition prevents tug-of-war between kinesin and dynein, but I do not see strong evidence supporting this view. Example kymographs in Figure 8a show little to no colocalization of kinesin with DDBE-oskar in the absence of Tm1. There is only infrequent and transient colocalization of kinesin with DDBE. I also do not see colocalization of plus end-directed kinesin (white arrowhead) with dynein. Most likely, this kinesin walks to the plus end on its own, independent of dynein.

During our revisions we identified an error in Figure 8c (now Figure 6c), in which the colors assigned to the 'Khc only' and 'Khc + dynein' fractions were reversed, giving the impression that the majority of plus-end-directed kinesins co-migrate with dynein signal in the absence of Tm1, which is not the case. As the reviewer correctly observes from the example kymograph, the majority of rare plus-end-directed Khc runs in the absence of Tm1 do not co-migrate with either dynein or oskar RNA. We have corrected the labeling and apologize for the error. Furthermore, we have updated the text in the manuscript to better reflect the infrequent nature of the runs demonstrated in Figure 8a (now Figure 6a) and their transient colocalization with other assay components (Lines 689-692).

We also acknowledge the need for precision in discussing a tug-of-war between dynein and kinesin-1, which suggests sustained opposition between active motors of opposite polarity that can result in a stalemate. While our data indicate that motor opposition can occur with RNA-linked DDBE-osk and Khc FL, and that Tm1 can reinforce Khc autoinhibition to regulate this process (see Figure 6e of the revised manuscript and response to the reviewer's next comment), the reviewer is correct to point out that the system does not display characteristics of a classical tug-of-war. We have therefore removed reference to a tug-of-war in this context and instead described our conclusions in more general terms of competition and opposition between dynein and Khc FL.

In the presence of Tm1, kinesin exhibits robust colocalization to DDBE-oskar and is transported by dynein to the minus end. These results show that Tm1 is the adapter that links kinesin-1 to the rest of the transport complex. However, they did not convincingly demonstrate that Tm1-dependent inhibition of kinesin is important for the rapid transport of the entire complex to the minus end. Given that full-length kinesin is autoinhibited on its own, it is unlikely to reduce the velocity of the complex even in the absence of Tm1-mediated inhibition. The authors should either tune down their conclusions based on these results or present more compelling experimental evidence that Tm1-mediated inhibition is important for the minus-end directed transport of kinesin.

We showed in our study that the isolated full length Khc does retain some activity in the absence of Tm1, demonstrating that autoinhibition is not complete. We also showed that association of full length Khc with RNPs reduces dynein's velocity in the minus-end

direction (by ~45%) when compared to experiments in which Tm1 was included and thus available to reinforce Khc inhibition (Figure 6e of the revised manuscript). This effect was not due to Tm1 regulating dynein directly, as Tm1 does not affect DDBE-osk velocity in the absence of Khc (Figure S12 of the revised manuscript). We believe these observations provide compelling evidence that dynein movement can be antagonized by full length Khc, even if Khc tends to be in an autoinhibited conformation, and that Tm1 suppresses this effect. This notion is in keeping with our observation that Tm1 reduces the frequency of rare plus-end-directed movements of Khc that can occur following a bout of minus-end-directed transport with dynein (Figure S15 and lines 392-394). Although the underlying mechanism for Khc-based antagonism of dynein is unclear, it could be related to stochastic engagement of Khc with the microtubule, thereby opposing dynein-driven movement (a possibility previously mentioned in the first submission and now included on lines 415-417 and 537-538 of the revision). We now appreciate that we should have made the points about Khc-mediated inhibition of dynein clearer and now do so on lines 413-419 and 537-539.

Whilst we believe that the evidence that Tm1-mediated inhibition of the full length Khc promotes rapid minus-end-directed transport of dual motor RNPs is compelling, we have erred on the side of caution with the associated language by using terms such as 'suggests...', 'indicates...', 'consistent with...', and 'supports a model in which...' in the sections highlighted above.

One possible experiment would be to repeat these assays using the Khc940 construct (if it still binds to Tm1 and/or RNA), which exhibits robust motility and is strongly inhibited by Tm1.

This is another very good suggestion. We have now performed experiments with the Khc940 motor and DDBE-osk in the presence and absence of Tm1. We find that, despite Khc940 exhibiting much more plus-end-directed motion than Khc FL in this system (as expected from the removal of the IAK motif), Tm1 still imparts an overall minus-end-directed bias to transport events of this motor. This finding, which is documented in Figure 6b, c and Figure S14 and lines 421-443, reinforces our idea that fully shutting down Khc activity involves a combination of inhibitory interactions involving the Khc regulatory tail (i.e. the Khc motor-IAK interaction and the action of Tm1 through the AMB domain), and that this process enables rapid transport of Khc-associated osk RNPs towards minus ends by DDBE. We are grateful to the reviewer for prompting us to do this informative experiment.

Minor concerns

1. Figure 7 is a control experiment and does not deserve to be a main text figure on its own. I recommend combining a part of this figure with Figure 8 and relegating its detailed analysis to the supplement.

The data presented in Figure 7 have now been integrated into Figure S12 of the revised manuscript, and Figure 8 (now Figure 6) has been expanded and supplemented by Figure S14 to include data from the new experiments with Khc940 in the presence of DDBE-osk.

2. The analysis of the data shown in Figure 8 can be slightly misleading as is (see major comment #3 above). Kinesin does not seem to localize robustly to oskar without Tm1. Therefore, the frequency of kinesin runs colocalizing with oskar in the presence and absence of Tm1 (as this seems to be the most profound effect) needs to be shown clearly before the more detailed analysis shown in this figure.

Inclusion of Tm1 in the assay with DDBE-osk and Khc FL has two profound and related effects on Khc motility. One effect, as highlighted by the reviewer, is an increase in association of Khc with oskar RNA and therefore with the DDBE machinery. The result of this increased association is the other profound effect, which is an increase in observed Khc motility on microtubules. We believe that the causal relationship between these effects requires their presentation together. We have, however, worked on improving the clarity of this part of the manuscript, which included experimenting with introducing the motility and co-localization effects in different orders. We have now come up with a formulation that we believe, together with correcting the unfortunate labeling error in Figure 8c of the original manuscript (as described in Major Comment 3 above), makes for a straightforward reflection of our observations and analyses. We have also strengthened the colocalization data in this panel by analyzing the rare plus-end-directed Khc runs in the presence of Tm1 (Figure 6c). This work provides further evidence that Tm1 couples Khc to DDBE-osk (line 398-402).

3. Lines 359-360: The authors wrote “This work involves the first reconstitution of a native cargo-adaptor-motor complex that recapitulates regulation of both dynein and kinesin motors.” I think this is an overstatement and needs to be softened. Recent studies on in vitro reconstitution of bidirectional transport machinery showed different types of motor regulation. For example, Fenton et al and Canty et al showed dynein transports kinesin to the minus end when both motors are recruited by the TRAK adaptor. I recommend authors briefly discuss these and other relevant studies that used a similar approach and more clearly state the differences between their system with the recent work in the field.

We believe our statement highlighted by the reviewer was accurate as the work by Fenton et al., Canty et al. and others on dual motor assemblies did not include the cargo and thus the regulatory role it may play. Nonetheless, we agree with the reviewer that we should do more to put our work in context of other studies that have reconstituted dynein-kinesin assemblies. We have now expanded on the findings of other studies (lines 526 - 531), including those of Fenton et al. and Canty et al. (both of which we had previously cited). We have also more clearly stated the differences between our system and previously reported systems, which has also led to the statement questioned by the reviewer being softened (line 531-533). We now write: ‘To our knowledge, Tm1 is the first example of a protein that integrates dynein and kinesin behavior in a reconstituted dual-motor assembly on an intact native cargo.’

Reviewer #2:

Remarks to the Author:

In this manuscript Heber and co-workers discuss how the relation between kinesin and dynein, two protein motors that transport RNA cargoes (among others) with opposite polarity in the microtubules system, is established. They focus in particular on the protein Tm1, a regulator of kinesin activity, and use in vitro functional and binding assays and an ensemble of methods reporting on interactions, structure and dynamics to show that Tm-1 binding is connected to a conformational change in the interacting region of kinesin, and that this strengthen kinesin inhibition. At the same time, the regulator links the inhibited kinesin to the dynein protein-RNA complex. The work provides an initial overview of the coupling of the two motors, both at the functional and molecular level.

This is an interesting paper that provides important novel molecular insight into a complex coupling mechanism essential to mRNA local translation. The study is clearly and logically structured and

the data are well presented. Conclusion are, in general, not overstated and the final model presented integrates effectively the insight gained in this study. Overall, this is a high quality paper.

There are a few issues that need to be addressed prior to publication – in particularly in the discussion of the structural data. This are important, as the mechanistic detail of the coupling is a key part of the paper.

Line 212. The discussion on short transversal relaxation times of the amino-terminal part of the Khc construct leading to the loss of the NMR signal is based, as I understand it, on the assumption that this region folds as a coil-coil. This is very likely, but should be validated, for example using CD.

We thank the reviewer for the suggestion. We have now performed the CD analysis of this Khc construct (855-975) and provide the results in the new Figure S8 and lines 245-247. The CD spectrum exhibits a signal typical of α -helix-rich proteins with a 222/208 nm ratio of 0.97, which is consistent with the presence of a coiled coil (Lau et al. 1984). However, as the crystal structure in Dimitrova et al., (2021) spans only residues 855-923, we cannot state with certainty that residues 924-936 are part of the coiled coil. It is also possible that these residues are flexible and undergo conformational exchange in the intermediate exchange regime, possibly due to bridging the coiled coil and the intrinsically disordered part of this construct.

Later on the page. The increase in linewidth in the N-terminal part of the resonance-assigned region of the kinesin upon Tm1 binding, with and without RNA (Figures 5c and d) seems monotonous.

This would be consistent with a conformational transition spreading from the N-terminus of the protein. It would be worth the authors comment on this. Again, this increase should be checked directly, using either CD or the carbon chemical shifts of the backbone resonances of the protein. In general, these chemical shifts should be discussed.

We thank the reviewer for their thorough examination of the NMR data. If the reviewer means by “monotonous” that the change in linewidth due to addition of Tm1 is independent of RNA, then this is correct (as stated in the manuscript, lines 254-255). We are not sure what the reviewer means by “...a conformational transition spreads from the N-terminus”. We stated that the linewidth increases for the region including the IAK motif as the peaks disappear upon addition of Tm1. As the reviewer points out, a change of conformation into a coiled coil is a possibility that we envisaged when we mentioned that these linewidth increases are consistent with an “induced conformational change” in the original manuscript. We have now expanded this section in the manuscript to clarify this issue (lines 263-270):

“Because signals from the affected Khc tails were broadened beyond detection, we could not assess secondary chemical shifts and thus whether the corresponding tail region undergoes a conformational change in the Tm1-bound state. Nonetheless, the intensity changes observed for the IAK motif and its proximal residues indicate either an induced conformational change or a second interaction of the Tm1 N-terminal domain with this region of the Khc tail. Therefore, although the IAK motif is not required for Tm1-mediated inhibition of Khc (Figure 2a-c and Table S1), its chemical environment is affected by Tm1 binding, suggesting that Tm1 modulates the interaction of the Khc motor with the tail.”

As suggested by the reviewer, we have now determined CD spectra for the Khc 855-975-Tm1 1-335 complex (Figure R2). However, this proved inconclusive, because both the

individual proteins and their complex are predominantly α -helical and CD is not accurate enough to determine such small changes in secondary structure as an extension of an α -helix by a few amino acids. In addition, the fact that addition of Tm1 to Khc adds α -helical content makes a direct comparison between bound and unbound Khc impossible.

With regards to secondary chemical shifts, we mentioned in the text:

“Backbone assignment experiments and resulting secondary chemical shifts revealed that the visible peaks are disordered and correspond to aa 937-975”.

This conclusion is based on the carbon chemical shifts ($C\alpha$ and $C\beta$) of free Khc 855-975 (all chemical shifts are submitted to the BMRB (accession code 51888), as stated in the manuscript). We have now added a secondary chemical shift plot to the supplement (Figure S7) and discuss this point in lines 237-239, as requested by the reviewer:

“Backbone assignment experiments and resulting secondary chemical shifts revealed that the visible peaks are disordered and correspond to aa 937-975, the part of the Khc tail that contains the auto-inhibitory IAK motif (Figure S7).”

We thank the reviewer for leading us to reevaluate this part of the study, as we noticed that the intensity of peaks between residues 937 to around 948 decreases by ~50 % and chemical shift perturbations are absent. This is consistent with one Tm1 binding to one tail of Khc and the other Khc tail remaining unbound, which would confirm the 1:2 binding observed previously by Dimitrova-Paternoga et al. 2021 and our mass photometry measurements (Figure S1). We now mention this point in lines 259-265. The remaining intensities should thus correspond to the unbound fraction of the Khc tail. As these peaks have no chemical shift perturbations, the conformation did not change and the secondary structure of the Khc tail remained unchanged. The bound fraction of the Khc tail may change conformation but corresponding intensities are broadened beyond detection and therefore we cannot make any statements about potential secondary structure formation based on secondary chemical shifts. We now make this point in lines 263-265.

“Interestingly, in the presence of Tm1 1-335, the intensities of the most strongly affected Khc resonances typically decreased by ~50 %. This indicates an effect of Tm1 on half the Khc tail molecules. Considering the established 2:1 stoichiometry of Khc:Tm1 (Figure S1, Ref. 43) we envisage Tm1 affecting only one chain of the Khc tail dimer. Because signals from the affected Khc tails were broadened beyond detection, we could not assess secondary chemical shifts and thus whether the corresponding tail region undergoes a conformational change in the Tm1-bound state.”

Figure R2: a Secondary structure content of Khc 855-975, Tm1 1-335 and the Khc 855-975-Tm1 1-335 complex calculated according to the CDSSTR, Selcon3 and Contin-LL methods using the Dichroweb server compared to predicted values of structural models. **b** Comparisons of experimental CD spectra of Khc 855-975, Tm1 1-335 and the Khc 855-975-Tm1 1-335 complex and theoretical CD spectra of their structural models predicted by PDBMD2CD (Drew and Janes 2020). Comparisons to theoretical CD spectra of structural models was performed using PDBMD2CD (Drew and Janes 2020). Models for Khc 855-975 and the Khc 855-975-Tm1 1-335 complex were derived from the models used for SAXS data analysis. A dimeric model for Tm1 1-335 was generated using AlphaFold2 (Jumper et al. 2021; Evans et al. 2021).

Also on this, mentioning the IAK motif in this context, may result confusing, as the TM1 binding and inhibition do not depend on the IAK.

We mention the IAK motif here because residues in this motif are affected by Tm1 binding, as shown in Figure 4. We have added a statement to the text (lines 268-270) that we think clarifies the situation:

“Therefore, although the IAK motif is not required for Tm1-mediated inhibition of Khc (Figure 2a-c and Table S1), its chemical environment is affected by Tm1 binding, suggesting that Tm1 modulates the interaction of the Khc motor with the tail.”

Line 236. The binding of the motor domain results in broadening and shifts that are not monotonous. This is consistent with the hypothesis discussed by the authors, i.e. that the binding sites of the motor region and of TM1 to the carboxyterminal part of the kinesin are proximal but largely non-overlapped. Notably, in Figure 5G, some of the shifts caused by the binding of the motor head (say residues 953-967) are the same whether TM1 is present or not. As I understand it, this is consistent with what discussed by the authors on the location of the binding sites of the two regulatory interactions and, it seems to me, confirms TM1 does not displace the motor head. On other residues instead, changes are observed which are more difficult to interpret, likely resulting by the two combined interactions.

We understand from this comment that the reviewer agrees with our interpretation of these data. Nonetheless, we have added a sentence to the text to attempt to improve the clarity (lines 283-285):

“However, Tm1 1-335 did lead to further intensity decreases of the resonances corresponding to the IAK motif and the flanking residues (Figure 4f, g), showing that Tm1 can bind to the Khc tail when it is associated with the motor domains.”

The reviewer is correct that additional chemical shift perturbations can be observed after addition of Tm1. However, they are very small and we would like to refrain from describing possible causes to avoid potential overinterpretation.

In general, the NMR data should be discussed more in detail. At the moment this detail is missing and the basis for the conclusions drawn are often implicit.

As described above, we have revised both the presentation and discussion of the NMR data in the manuscript in order to provide more detail and clarify the basis for our conclusions.

Line 257 and figures 6. We need more detailed information of the number and type of models. It is not very easy to judge whether this is one of several models that are compatible with the data or the only one. This should be discussed explicitly and more in detail in the paper.

Detailed information on numbers and types of the models used was provided in the Materials & Methods section and the legends to Figures 6 and S8 in the initial submission. To make the information more accessible, we now also provide it in the main text (lines 309-313 and 322-324). We thank the reviewer for pointing out this issue.

“We subsequently randomized backbone angles in the connecting linker and tail regions of Khc to create a large pool of 5000 possible structures for both Khc and Khc-Tm1. In a second step, the C-terminal tail of one Khc chain was pulled into the binding pocket of the motor domain resulting in a second pool of 5000 structures for both Khc and Khc-Tm1 with random orientation of the stalk region and the IAK peptide bound to the motor domain dimer.”

“Examining the angles of consecutive coiled-coil regions in an ensemble of well-fitting Khc models (five best models are shown in Figure S10d),...”

Even better would be some validation of the model.

SAXS yields extensive information on the conformational state of protein complexes, including those with a high degree of flexibility, but lacks sufficient resolution for mutagenesis-based validation. We have therefore attempted to validate our models by collecting SAXS data for the Khc mutants described in the manuscript. Unfortunately, these mutants were prone to oligomerization in the conditions required for SAXS measurements leaving us unable to acquire interpretable results. In response to Reviewer 3's request to use models similar to those recently described by Weijman et al. 2022 and Tan et al. 2023, we repeated our modeling with new pools consisting of structures with more rigid stalk arrangements, in which the hinge2 region is not flexible. The results from fitting these models to our SAXS data (presented in Figure S11) validate our original finding that Tm1-binding results in compaction of Khc. Further validation of these models using high-resolution methods will be an important part of future studies. However, we anticipate that this will be very challenging as several groups have tried and failed to achieve atomic resolution structures of full length kinesin-1.

When discussing the microtubules experiments, I find that the authors could explain more in detail how they interpret the images we are looking at. For example, in page 4 line 93 they mention Figure 1C shows transport of oskar RNA. One can imagine how transport is derived, but it would be useful to explain it.

We apologize for the lack of clarity in how information is extracted from the kymographs presented in Figure 1c and elsewhere in the paper. We have now added a description of how these are interpreted to the legend of Figure 1c and expanded the 'In vitro single-molecule reconstitution assays, TIRF microscopy and image analysis' subsection of the Materials and Methods (lines 856-861) to explicitly define the metrics that are derived from kymograph analysis.

"Kymographs (time vs. distance plots projecting the position of fluorescently labeled particles over time along an individual microtubule; moving particles have a horizontal component whilst static particles are projected as vertical lines)..."

"Binding events were defined as those that persisted for a minimum of ~2.5 s. Motile events were defined as binding events in which unidirectional movement of fluorescently-labeled assay components could be observed for a minimum of 5 pixels on the position axis (x-axis) of the kymograph. Motor run lengths and average velocities were extracted from kymographs by measuring the total observed displacement on the position axis (x-axis) or the inverse of its average slope, respectively."

Just below this sentence, the authors mention the value does not change significantly and give values of 4.8% and 2.8 % for the experiment in the absence and in the presence of Tm1. It would be probably better to have also the errors here, or avoid specific numbers.

We appreciate the reviewer pointing this out and agree that the nature of these metrics and the statistical tests used to compare them should be made clearer. Due to their paucity, these percentages are derived from the total number of observations rather than from repeated measures; they therefore do not have any error associated with them. The datasets can nonetheless be statistically compared by Fisher's exact test, which shows that they are not significantly different. We have updated the text on lines 127-130 to state our use of this statistical test, and included data from additional replicates of the

experiment, thereby increasing the number of complexes analyzed, and thus the confidence in the conclusion:

“In the absence of Tm1, we observed occasional co-migration of osk with Khc (47/748 complexes (6.3 %); Figure 1c). This frequency did not change significantly in the presence of Tm1 (9/205 complexes (4.2 %); $p=0.4016$ compared to in the absence of Tm1 (Fisher’s exact test); Figure 1c),...”

Reviewer #3:

Remarks to the Author:

This is an interesting paper showing that tropomyosin-1 links kinesin-1 in an autoinhibited state to a dynein-containing osk mRNA transport complex. The idea is that kinesin is loaded already in the nurse cells and only gets active when arriving in the oocyte where kinesin-1 takes over the transport. This is a novel and interesting concept, which would be of significant interest to the cytoskeleton and motor field.

However, I have some issues with the robustness of the science. The numbers of observations are generally very low and seem to have been derived in many cases from single experiments, often less than 10 microtubules that could possibly have been found in a single field of view. Therefore, the statistical significance is meaningless as a pipetting error alone would be enough to result in a change in binding events.

We thank the reviewer for making this important point. Throughout this study, our conclusions were drawn from experiments that had been independently repeated and reproduced several times, albeit not always under identical conditions (hence the exclusion of some replicates from the manuscript). Each independent experiment used a new protein assembly and was typically performed on a different day. In addition, each experiment typically involved several independent injections of protein assemblies into imaging chambers. This information has now been added to the Materials and Methods (line 864-869). We have now performed several further independent repeats of experiments using our standard experimental conditions to increase confidence in the conclusions. Details of how many individual motor complexes, microtubules, imaging chambers per experiment, and independent experiments were analyzed are now included in all figure legends, as well as in Tables S1-S4. Furthermore, as requested below by the reviewer, we now present the means/medians of independent experimental repeats within each figure, in addition to the distribution of individual data points.

Some of the kymographs provided also seem not to support the quantitative data. The velocity of KHC-containing events shown in Figure 8a seems lower than that of dynein without KHC. However, the quantification in 8d and 7b contradict this. Of course, if presence of KHC would lower the speed of dynein, this would suggest that loaded KHC is able to bind to microtubules and not being autoinhibited. Thus it is fairly important for the conclusions of the manuscript to solidify those findings.

Selection of kymographs in which a small number of complexes faithfully reflect the behavior of hundreds or thousands of complexes in a dataset is a common challenge for single-molecule studies, highlighting the importance of robust quantification. This challenge is compounded when quantification of the entire dataset reveals multiple changes in motor behavior, as is the case when Tm1 is included in our assays. Nonetheless, we believed that the kymographs presented in Figure 8a (now Figure 6a)

achieved this and appropriately illustrated the various changes in Khc FL behavior that occur when complexed with DDBE-osk in the presence of Tm1.

On the particular point of perceived differences in dynein velocity when bound to Khc FL, we assume the reviewer is comparing the kymograph showing dynein signal in the presence of Tm1 to the kymograph of the corresponding Khc FL channel. Whilst we did not get the impression that the Khc-associated dyneins move more slowly than the Khc-free dyneins in these kymographs when constructing the figure, we nonetheless wanted to determine if this is the case. We therefore analyzed the motility in the kymographs and found that in the presence of Tm1, the velocity of Khc-bound dynein is not significantly lower than that of Khc-independent dynein ($0.333 \pm 0.305 \mu\text{m s}^{-1}$ (N=43 complexes) vs $0.389 \pm 0.275 \mu\text{m s}^{-1}$ (N=24 complexes), respectively (mean \pm SD; $p=0.4552$ from t-test)). Thus, we do not believe there is any contradiction between these kymographs and our model for how Tm1 affects Khc.

Regarding the point about small differences in measured velocities between Figures 7b and 8d (now Figures S12b and 6e, respectively), we believe this particular comparison is not appropriate. While it is true that the measured velocities of DDBE-osk in the absence of Tm1 and Khc in Figure S12b are marginally lower than those for DDBE-osk in the presence of these factors in Figure 6e, the experiments for these two panels were performed months apart from each other with different Egl-BicD preparations and the measured velocities are within our historical expectations for variability under these circumstances. Thus, we continue to believe that the most appropriate comparison of velocity is that presented in the paper, in which values for minus-end-directed Khc in the presence and absence of Tm1 are compared in parallel experiments across multiple independent replicates. These analyses reveal that association of Khc FL with dynein in the absence of Tm1 reduces dynein velocity by ~45% compared to when Tm1 is absent, suggesting that Khc FL does indeed oppose dynein activity and that Tm1 limits this opposition (a conclusion we draw on lines 415-417).

Two recent manuscripts updated our knowledge on kinesin-1 autoinhibition: 10.1126/sciadv.abp9660 and 10.1101/2023.01.26.525761. The former is briefly mentioned, the latter not. It would be important that models compared to SAXS data here would include models based on those recent findings. The data provided in the other papers show a compact shape of KHC in the absence of TM-1. Therefore a detailed discussion why Tm-1 is required to compact kinesin-1 in the current study is absolutely required.

Indeed, these two manuscripts provide important knowledge about kinesin-1 autoinhibition. We had made reference to both in our original discussion (lines 381-383 and 400-403 of the original manuscript) but now discuss them in more depth following prompting by this reviewer and Reviewer 2.

In the two other studies, the compact conformation of the mammalian kinesin was isolated by size exclusion chromatography and static molecules examined by cryoEM. Tan et al. additionally stabilized the compact conformation by chemical crosslinking. Considering their approaches, it is not surprising that the two studies describe a compact, closed Khc conformation. In contrast, we use NMR and SAXS to investigate dynamics and conformational flexibility of the kinesin structure in solution (we now highlight the different methods used in these studies and their potential significance on lines 507-518. We find that binding of Tm1-bound Khc forms a more compact particle in solution, leading us to propose that Tm1 stabilizes the autoinhibited conformation of Khc.

Common to the Tan and Weijman studies is the rigid connection of cc1 and cc2 (via hinge2) generating a colinear arrangement of the two helix axis, which results in a topology of the stalk that resembles our Khc-Tm1 models and is quite different from our Khc-only models that have a sharp turn of the linker between cc1 and cc2. We tried to confirm the rigid arrangement of the hinge2 region (aa 542-646) experimentally, but our efforts to purify soluble, well-behaved constructs containing the region were not successful. In the absence of experimental evidence for this arrangement, we decided to keep the linker structure flexible during the modeling.

To address the reviewer's comment, we generated models of Khc and Khc-Tm1 with a rigid cc1-cc2 region (as predicted by AlphaFold2) that is very similar to the arrangement of this region in the structures of the recently published models. The resulting structures that best fit the SAXS data for Khc-Tm1 have a topology and χ^2 values comparable to those of the structures obtained by the original model described in our manuscript, which showed a similar, compact arrangement with folding of the stalk at hinge3, as described in both other studies (Weijman et al., 2022, Tan et al., 2023). In contrast the best fitting Khc model exhibits a χ^2 increase of 15 % compared to the best fitting model in our manuscript (3.54 compared to 3.07), showing a less good fit. In all of the best fitting new models, the stalk exhibits a larger distance between the cc3-cc4 linker region and cc1-cc2 and thus, similar to our original models, a triangular stalk arrangement, rather than a compact packing of cc3 and cc4 onto cc1-cc2. Khc models with more compact packing show $\chi^2 > 5$. Thus, we obtain similar results with both pools of models: we find a more open conformation for unbound Khc and compaction of Khc upon Tm1-binding. We now show the new results in Figure S11 and describe them in lines 334-350 of the manuscript. Since the available preprint from Tan et al. does not contain the residue specific data for cross-links within the stalk region, the sequence of the protein in their study is different, and there are no pdb files published, we could not generate models that include these data. However, we can conclude that the compact arrangement does not fit our SAXS data with Khc, and that a rigid straight arrangement of cc1-cc2 is less compatible than a tight turn. We stress in the manuscript that our fits cannot account for alternative Khc conformations (both autoinhibited and open) that may exist as part of dynamic conformational equilibria, but the good agreement of individual models with experimental SAXS data suggests that they may represent a predominant conformation of the proteins of the structural ensembles in solution. We therefore would rather not extend the discussion in the manuscript with alternative modeling suggestions.

Therefore, I can only recommend publication after experiments have been conducted with suitable independent repeats and data are provided so that variation within each experiment and across repeats are presented. I would like to refer the authors to this for guidance:
10.1083/jcb.202001064

As described above, we have taken multiple steps to address the reviewer's concerns about independent repeats and data presentation in response to their very helpful suggestion. This has included presenting variation in mean values across repeats in the figures, as requested. The mean values for individual experiments are also color-coded to enable the reader to evaluate differences within each experiment (note that we have not colored individual data points based on which experiment they came from as this made the plots utterly incomprehensible).

There is often significant variation between repeats within an experimental condition for metrics that are expressed as averages per microtubule because some replicates had a small number of microtubules that could be analyzed (either because few microtubules

bound the surface or because microtubule overlaps meant that associated particles could not be faithfully tracked (an issue now mentioned in the Materials and Methods (lines 862-864)). This issue can be exacerbated for cases in which Khc is inhibited because there is often a very low number of Khc complexes per microtubule, which increases variability of 'per microtubule' means per experiment. We therefore believe that averaging the means of independent experiments would obscure the overall trends that we observe. Moreover, statistical demonstration of these trends using per experiments averages would require an impractically large number of experimental replicates. For this reason, and for the sake of consistency throughout the paper, we have elected to evaluate statistical significance in our experiments using pooled values of individual measurements (i.e. individual microtubules or individual motor complexes) from all experimental replicates within a condition. This practice is widespread in the field of single-molecule motor assays due to the low-throughput nature of the experiments, and is explicitly stated in each figure legend. Nonetheless, we have not drawn firm conclusions unless they are also supported by inspecting the parallel per experiment means.

Decision Letter, first revision:

Message: Our ref: NSMB-A47478A

24th Oct 2023

Dear Dr. Ephrussi,

Thank you for submitting your revised manuscript "Tropomyosin 1-I/C co-ordinates kinesin-1 and dynein motors during oskar mRNA transport" (NSMB-A47478A). It has now been seen by the original referees and their comments are below. The reviewers find that the paper has improved in revision, and therefore we'll be happy to accept it in principle in Nature Structural & Molecular Biology, pending minor revisions to satisfy the referees' final requests and to comply with our editorial and formatting guidelines.

To facilitate our work at this stage, it is important that we have a copy of the main text as a word file. If you could please send along a word version of this file as soon as possible, we would greatly appreciate it; please make sure to copy the NSMB account (cc'ed above).

Sincerely,

Dimitris Typas
Associate Editor
Nature Structural & Molecular Biology
ORCID: 0000-0002-8737-1319

Reviewer #1 (Remarks to the Author):

The reviewers adequately addressed my concerns. I support the publication of this work in NSMB.

Reviewer #2 (Remarks to the Author):

I am looking forward to see this interesting and well-drafted paper published.

Reviewer #3 (Remarks to the Author):

I am satisfied that the authors have adequately addressed all my concerns/suggestions and I am happy to recommend publication.

Final Decision Letter:

Message 2nd Jan 2024

:

Dear Dr. Ephrussi,

We are now happy to accept your revised paper "Tropomyosin 1-I/C co-ordinates kinesin-1 and dynein motors during oskar mRNA transport" for publication as an Article in Nature Structural & Molecular Biology.

As soon as your article is published, you can generate your shareable link by entering the DOI of your article here: http://authors.springernature.com/share. Corresponding authors will also receive an automated email with the shareable link

Your paper will be published online soon after we receive proof corrections and will appear

in print in the next available issue. You can find out your date of online publication by contacting the production team shortly after sending your proof corrections.

You may wish to make your media relations office aware of your accepted publication, in case they consider it appropriate to organize some internal or external publicity. Once your paper has been scheduled you will receive an email confirming the publication details. This is normally 3-4 working days in advance of publication. If you need additional notice of the date and time of publication, please let the production team know when you receive the proof of your article to ensure there is sufficient time to coordinate. Further information on our embargo policies can be found here:

<https://www.nature.com/authors/policies/embargo.html>

Please note that *Nature Structural & Molecular Biology* is a Transformative Journal (TJ). Authors may publish their research with us through the traditional subscription access route or make their paper immediately open access through payment of an article-processing charge (APC). Authors will not be required to make a final decision about access to their article until it has been accepted. [Find out more about Transformative Journals](https://www.springernature.com/gp/open-research/transformative-journals)

Authors may need to take specific actions to achieve [compliance with funder and institutional open access mandates](https://www.springernature.com/gp/open-research/funding/policy-compliance-faqs). If your research is supported by a funder that requires immediate open access (e.g. according to [Plan S principles](https://www.springernature.com/gp/open-research/plan-s-compliance)) then you should select the gold OA route, and we will direct you to the compliant route where possible. For authors selecting the subscription publication route, the journal's standard licensing terms will need to be accepted, including [11](https://www.springernature.com/gp/open-research/policies/journal-

self-archiving policies. Those licensing terms will supersede any other terms that the author or any third party may assert apply to any version of the manuscript.

Sincerely,

Dimitris Typas
Associate Editor
Nature Structural & Molecular Biology
ORCID: 0000-0002-8737-1319